

# Contributions of different anthropogenic volatile organic compound sources to ozone formation at a receptor site in the Pearl River Delta region and its policy implications

Zhuoran He[1,2], Xuemei Wang[3], Zhenhao Ling[1,2*], Jun Zhao[1,2], Hai Guo[4], Min Shao[3], Zhe Wang[4,*]

[1]School of Atmospheric Sciences, Sun Yat-sen University, Guangzhou, China

[2]Guangdong Province Key Laboratory for Climate Change and Natural Disaster Studies, Sun Yat-sen University, Guangzhou, China

[3] Institute for Environmental and Climate Research, Jinan University, Guangzhou, China

[4] Department of Civil and Environmental Engineering, Hong Kong Polytechnic University, Hong Kong, China

*Correspondence to*: lingzhh3@mail.sysu.edu.cn (Zhenhao Ling); z.wang@polyu.edu.hk (Zhe Wang)

**Abstract.** Volatile organic compounds (VOCs) are key precursors of photochemical smog. Quantitatively evaluating the contributions of VOCs sources to ozone ($O_3$) formation could provide valuable information for emissions control and photochemical pollution abatement. This study analysed the continuously measured VOCs during the photochemical season in 2014 at a receptor site (Heshan site, HS) in the Pearl River Delta (PRD) region, where photochemical pollution has been a long-standing issue. The averaged mixing ratio of measure VOCs was $34 \pm 3$ ppbv, with the largest contribution from alkanes ($17 \pm 2$ ppbv, 49%), followed by aromatics, alkenes, and acetylene. The positive matrix factorization (PMF) model was applied to resolve the anthropogenic sources of VOCs, coupled with a photochemical-aged-based parameterization that better considers the photochemical processing effects. Four anthropogenic emission sources were identified and quantified, with gasoline vehicular emission as the most significant contributor to the observed VOCs, followed by diesel vehicular emissions, biomass burning, and solvent usage. The $O_3$ photochemical formation regime at HS was identified as VOCs-limited by a photochemical box model with the master chemical mechanism (PBM-MCM). The PBM-MCM model results also suggested that vehicular emission was the most important source to the $O_3$ formation, followed by biomass burning and solvent usage. Sensitivity analysis indicated that in order to prevent the increment





of O₃ concentration, the abatement ratios of the individual VOC source vs. NOₓ should be higher than
3.8, 4.6, 4.6, and 3.3, respectively, for diesel vehicular emission, solvent usage, biomass burning, and
gasoline vehicular emission, respectively. Based on the above results, a brief review on the policies on
the controlling of vehicular emissions and biomass burning in the PRD region from a regional perspective
were also provided in this study. It reveals that different policies have been/being implemented and
formulated could help to alleviate the photochemical pollution in the PRD. Nevertheless, evaluation on
the cost-benefit of each policy is still needed to improve the air quality.
Key words: Anthropogenic emissions; Ozone formation; Pearl River Delta region; Policy implications
**1 Introduction**
Atmospheric volatile organic compounds (VOCs) have significant impact on air quality. Due to their high
chemical activity, VOCs are key precursors of ozone (O₃) and secondary organic aerosol (SOA). In
addition, some VOCs and their oxidation products are harmful to human health, which further deteriorates
air quality (Seinfeld and Pandis, 2006; ATSDR, 2007; Huang et al., 2014). VOCs have a variety of natural
and anthropogenic sources, including biogenic emissions and emissions from human activities (*i.e.,* fuel
and biomass combustion, fuel evaporation, solvent usage, industrial processes, etc.). It is relatively well
known that the two key ozone precursors (VOCs and NOₓ) synergize complex, nonlinear effects on ozone
formation. For a given region, depending on which precursor is the limiting factor for controlling ozone
formation, the ozone isopleth diagram (the mixing ratios of VOCs and NOₓ as two coordinates) can be
classified into VOCs and NOₓ limited regimes. In the VOCs-limited regime, the effective measure for
reducing ozone production is to control the VOC emissions and vice versa for the NOₓ-limited regime
(Jenkin and Clemitshaw, 2000).
In recent years, with rapid urbanization and industrialization, high O₃ mixing ratios were frequently
observed in the Pearl River Delta region (e.g., Zheng et al., 2010; Li et al., 2014; Wang et al., 2017).





Many previous studies have shown that photochemical $O_3$ formation was generally VOCs-limited in the
PRD region, and suggested that the reduction of VOC emission could effectively alleviate photochemical
$O_3$ formation (Guo et al., 2017). Therefore, source identification and quantification of VOCs are
prerequisites for the formulation and implementation of the most effective control measures of
photochemical pollution in the PRD region. Indeed, many efforts have been made to perform the source
apportionments of VOCs in this region by using different methods, including tunnel measurements,
receptor models, emission-based measurements, and emission inventory. Ho et al. (2009) quantified the
emission factors of 92 VOCs from gasoline, diesel, and LPG vehicles from a tunnel study in Hong Kong.
Guo et al. (2011b) and Zheng et al. (2013) characterized the source profiles of VOCs emitted from
industrial and vehicular sectors through samples collected directly from the plumes of the above sources.
These emission-based measurements provided clear attributions of VOCs from different sources and
emission factors for the emission-inventory to estimate the total amount of VOCs emitted from those
sources. In particular, Zheng et al. (2009) and Ou et al. (2015b) established a specific VOC emission
inventory to estimate the abundance of VOCs and to provide input data for different air quality models.
The inventory was applied to quantify the strength of vehicular emissions, solvent usage, and biogenic
emissions in the PRD.
In contrast to emission inventory, which estimates the emission strength based on emission factors and
emission activity, receptor models are useful for source apportionment of VOCs without any prior
knowledge of the emissions. As a widely used receptor model, positive matrix factorization (PMF) has
been employed in the source apportionment of VOCs in the PRD region (Guo et al., 2011a; Ling et al.,
2011; Lau et al., 2010; Ou et al., 2015a). For example, Ling et al. (2011) identified 10 sources of VOCs
at a receptor site in the PRD region and concluded that solvent usage and vehicular exhaust were the most
significant sources with average contributions of 51% and 37%, respectively. Results from source
apportionment using the PMF model demonstrated the important roles of vehicular emissions in ambient
VOCs in urban ($65 \pm 36\%$) and suburban ($50 \pm 28\%$ and $53 \pm 41\%$) environments of Hong Kong (Lau et





al., 2010; Ou et al., 2015a). However, uncertainties existed in the PMF analysis due to the assumption of
mass conservation during the transport of pollutants from emissions to the receptor site. To investigate
the influence of photochemical processes on the factorization of VOCs by the PMF model, Yuan et al.
(2012b) applied a photochemical age-based parameterization method to analyze the measured VOC data
at an urban site in Beijing. They found that the PMF-resolved factors were influenced by VOCs from a
common source at different stages of the photochemical processing, thus the independent source could
not be clearly identified. The results further suggested that when using the PMF model for VOC source
apportionments, it is necessary to assess if photochemical processing could influence the source
signatures of VOCs at the receptor site. Although many previous studies reported source apportionment
of VOCs in the PRD region using the PMF model, they did not take the influence of photochemical
processing into account, leading possibly to uncertainties in identifying and quantifying the source
attribution of VOCs.
In this study, the PMF model coupled with a photochemical-age-based parameterization method was
applied to the continuous real-time VOCs data from an intensive field campaign at a receptor site in PRD.
The model provides a more detailed and accurate description of the source characteristics of VOCs in the
PRD region. Furthermore, the contribution of different sources of VOCs to the photochemical $O_3$
formation and the sensitivity of ozone and its precursors were evaluated through a photochemical box
model coupled with the master chemical mechanism. Our results could provide valuable information,
facilitating local and regional policy-makers for proposing appropriate strategies and effective control
measures of VOCs and photochemical pollution.



## 2 Methodology

### 2.1 Measurements

The field measurement was conducted at the Heshan (HS) Atmospheric Supersite (22.728°N, 112.929°E, at an altitude of 60 m) in the western PRD region from October 22 to November 20, 2014. Figure 1 shows the surrounding environment at the sampling site. A detailed description of the Heshan site can be found in previous studies (Zhou et al., 2013, 2014). Briefly, the site is located in a rural area of the PRD region, about 50-80 km northeast of the urban central cities (i.e., Guangzhou and Foshan) of the PRD region. In addition to local emissions, the abundances of air pollutants at the HS during autumn and winter seasons are frequently affected by the outflow of air masses from the central cities, thus this site can be used as a representative of regional emissions in the PRD region (Zhou et al., 2014).

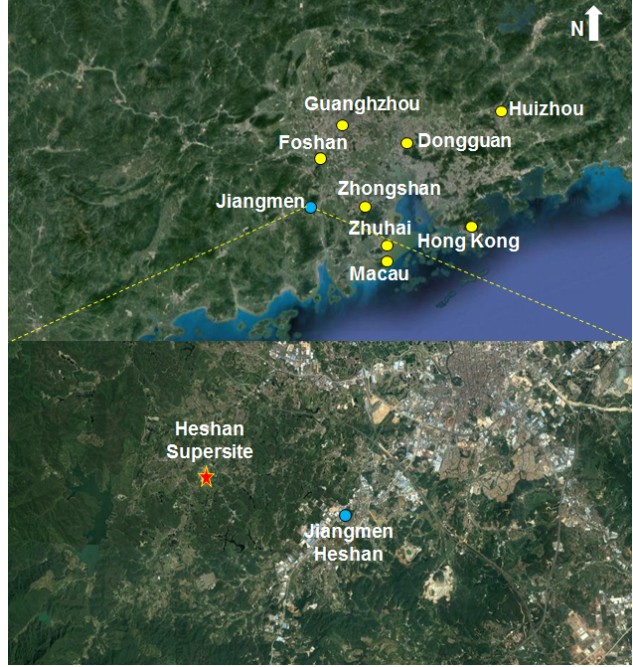

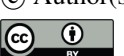



Figure 1. The sampling site and its surrounding environment in the Pearl River Delta region (top panel: overview of the
site location; bottom panel: zoomed-in view, red star denotes the site location)

An automated online gas chromatography-flame ionization detector (GC-FID) system measured hourly
concentrations of 58 VOC species at the site from October 22 to November 20, 2014. Detailed
descriptions of the configuration of the GC-FID system, the detection limits, and precision of VOCs can
be found elsewhere (Wang et al., 2008; Zhang et al., 2008a; Ling et al., 2017). Air-quality related trace
gases, including $O_3$, $NO$-$NO_2$-$NO_x$, $CO$, and $SO_2$, together with meteorological data, *i.e.,* temperature,
solar radiation, precipitation, relative humidity, wind speed, and wind direction were continuously
measured by the Guangdong Environmental Monitoring Center.
**2.2 Positive matrix factorization (PMF) model**
The Positive Matrix Factorization (PMF) (US Environmental Protection Agency (USEPA) version 5.0)
model was applied to the observed data for source apportionments of the VOCs. The PMF model is a
multivariate factor analysis tool that decomposes a matrix of speciated sample data into two matrices,
factor contributions, and factor profiles, which can be interpreted by an analyst as to explore the source
types and their contributions based on the measured data at the receptor site (Paatero and Tapper, 1994;
Paatero, 1997). It could be simplified as
$$x_{ij} = \sum_{k=1}^{p} g_{ik} f_{kj} + e_{ij},  \qquad (1)$$
where $x_{ij}$ is the $j$th species concentration measured in the $i$th sample, $g_{ik}$ the species contribution of the
$k$th source to the $i$th sample, $f_{kj}$ the $j$th species fraction from the $k$th source, $e_{ij}$ the residual for each
species, and $p$ the total number of independent sources (Paatero, 1997). The model could provide the
number of emission sources ($p$) and the distributed profiles ($f$) of each species in the individual source
after simulation.





The description of the model input was provided elsewhere (Guo et al., 2011a). In this study, the selection
of species for the model input followed several criteria: 1) The chosen species had relatively high
concentrations and/or were typical tracers for specific emissions. 2) Species with low abundance and/or
high uncertainties were excluded, *i.e.*, *cis*-2-pentene, diphenyl methane, and 1,3-diethylbenzene, because
more than a quarter of the samples for those species were below the detection limit. 3) Species related to
biogenic emissions, i.e., isoprene and *α/β*-pinene, were excluded as this study focused on the source
characteristics of anthropogenic emissions in the PRD region (Fuentes et al., 1996; Sanadze, 2004). A
total of 49 species, including 47 VOCs, MTBE (methyl tert-butyl ether), and acetonitrile (ACN) were
selected for the input data.
For the PMF modeling run, different numbers of factors were tested and an optimal number of factors
was determined based on both a good fit to the data and the most meaningful results. The uncertainty for
each species was determined to be sum of 10% of the VOC concentration and two times the detection
limit of the species (Paatero, 2000a; Lau et al., 2010). Concentrations below the detection limit were
replaced with half of the detection limit and their uncertainties were set to be 5/6 of the detection limit.
Missing concentrations were replaced by the geometric mean of the measured values and their
corresponding uncertainties were set to be four times the geometric mean values (Paatero, 2000b).
**2.3 PBM-MCM model**
The photochemical box model coupled with the master chemical mechanism (PBM-MCM) was applied
to quantify the contributions of VOC emission sources to in-situ $O_3$ formation. The PBM-MCM model
uses the concentrations of VOCs and trace gases, and the meteorological data as input to simulate the
total amount of photochemical $O_3$ formation at the site based on the master chemical mechanism (version
3.2), which consists of 5900 chemical species and 16500 reactions. Note that the physical processes,
including horizontal and vertical transport, were not considered in the model. Details of the model setup
and configuration can be found in previous studies (Saunders et al., 2003; Lam et al., 2013). In this study,



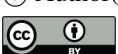

the hourly data of VOCs, five trace gases (i.e., $O_3$, NO, $NO_2$, CO, and $SO_2$) and two meteorological
parameters (*i.e.,* temperature and relative humidity) measured during the campaign were used as model
input.
In addition, the PBM-MCM model can be used to assess the sensitivity of $O_3$ photochemical production
to the changes in the concentrations of its precursors by calculating the relative incremental reactivity
(RIR) without a detailed or accurate knowledge of these emissions (Carter and Atkinson, 1989; Cardelino
and Chameides, 1995). The RIR is defined as the percent change in $O_3$ production per percent change in
the precursors. The RIR of a specific precursor $X$ at site $S$ is given by
$$RIR^S(X) = \frac{\left[P^S_{O_3-NO}(X) - P^S_{O_3-NO}(X-\Delta X)\right]/P^S_{O_3-NO}(X)}{\Delta S(X)/S(X)} \tag{2}$$
where $S(X)$ represents the measured concentration of precursor $X$, including the amounts emitted at the
site and those transported to the site and $\Delta X$ is the change in the concentration of precursor $X$ caused by
a hypothetical change $\Delta S(X)$ (10% $S(X)$ in this study). Here, $P^S_{O_3-NO}$ represents the $O_3$ formation
potential, which is the net $O_3$ production and NO consumed during the evaluation period and can be
calculated by the output from the PBM-MCM. A large positive RIR value of a specific precursor suggests
that the $O_3$ production could be decreased significantly if the emissions of this precursor were controlled.
In addition, the average RIR function of precursor $X$ can be calculated from
$$\overline{RIR}(X) = \frac{\sum_1^N[RIR^S(X)P^S_{O_3-NO}(X)]}{\sum_1^N P^S_{O_3-NO}(X)} \tag{3}$$
where $N$ means the number of days simulated.
In addition, considering both the reactivity and abundance of VOC species in different sources, the
relative contributions of the precursor $X$ can be calculated by (Ling et al., 2011; Ling and Guo, 2014).
$$Contribution(X) = \frac{\overline{RIR}(X) \times conc(X)}{\sum[\overline{RIR}(X) \times conc(X)]} \times 100\% \tag{4}$$
where $conc(X)$ was obtained from the measurement and PMF resolutions.





## 3 Results and discussion

### 3.1 General statistics

Figure 2 shows the time series of $O_3$ and total VOCs (TVOCs), as well as meteorological parameters (i.e., temperature, relative humidity) observed at Heshan site from 22 October to 20 November. It was found that two major episodes of high $O_3$ mixing ratios (maximum hourly-averaged mixing ratio > 100 ppbv, China II Standard) appeared during 24 October ~ 01 November and 13~19 November, respectively. Consistent with higher $O_3$ levels, the mixing ratios of TVOCs in $O_3$ episode days were higher, with the average values of 38 ± 3 ppbv and 30 ± 2 ppbv (mean ± 95% intervals) observed during $O_3$ episode and non-episode days, respectively, indicating that $O_3$ formation at Heshan site was probably VOC-limited. The measured 58 VOC species included 30 alkanes, 10 alkenes, 17 aromatics, and acetylene. Table 1 summarizes the average mixing ratios of the major VOC groups measured at the site from 22 October to 20 November, 2014. The average mixing ratio of total VOCs was 34 ± 3 ppbv, with the highest contributions from alkanes (17 ± 2 ppbv, 49%), followed by aromatics (9 ± 1 ppbv, 26%), alkenes (5 ± 1 ppbv, 15 %) and acetylene (3 ± 1 ppbv, 10%). This is consistent with previous measurements in this region (Guo et al., 2011a; Yuan et al., 2012a; Zou et al., 2015). The VOC mixing ratio at the Heshan site was similar to that in urban Shanghai and Beijing, with a range of 30.3-38.7 ppbv (Geng et al., 2009; Cai et al., 2010) and 29.4-43.4 ppbv (Song et al., 2007; Duan et al., 2008; Shao et al., 2009; Li et al., 2015), respectively. However, it was much higher than that in background areas of the North China Plain region, Yangtze River Delta region, and PRD (< 20 ppbv) (Tang et al., 2009; Yuan et al., 2012b; Zhu et al., 2016). The most abundant VOC species was ethane (3.86 ± 0.10 ppbv), followed by toluene (3.74 ± 0.22 ppbv), acetylene (3.42 ± 0.17 ppbv), propane (3.01 ± 0.14 ppbv), and ethene (2.94 ± 0.25 ppbv). The composition and the variations of VOCs suggest that the air masses arriving at the Heshan site may be through photochemical processing. In addition, incomplete combustion was likely to be the dominant source of VOCs at this site (Yuan et al., 2012a; Zhang et al., 2014; Yang et al., 2017), because typical tracers of

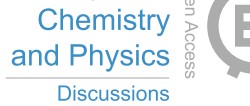



incomplete combustion (i.e., ethane, acetylene, ethene, and propane) were present with high

concentrations due to relatively lower photochemical reactivity of alkane (*i.e.,* ethane and propane) than

other VOCs. Indeed, the average diurnal variations of VOCs (Figure 3) presented relative higher mixing
ratios during the early morning and from the evening to midnight, which may be related to elevated traffic
emissions during rush hours and the constrained mixing height (Yuan et al., 2009). On the other hand, the
mixing ratios of VOCs started to decrease at 0900 LT (local time) and presented a broad trough during
daytime hours (0900-1900 LT), which were probably due to strong photochemical reactions, increased
mixing height and/or less VOC emissions (Yuan et al., 2009; Lau et al., 2010).

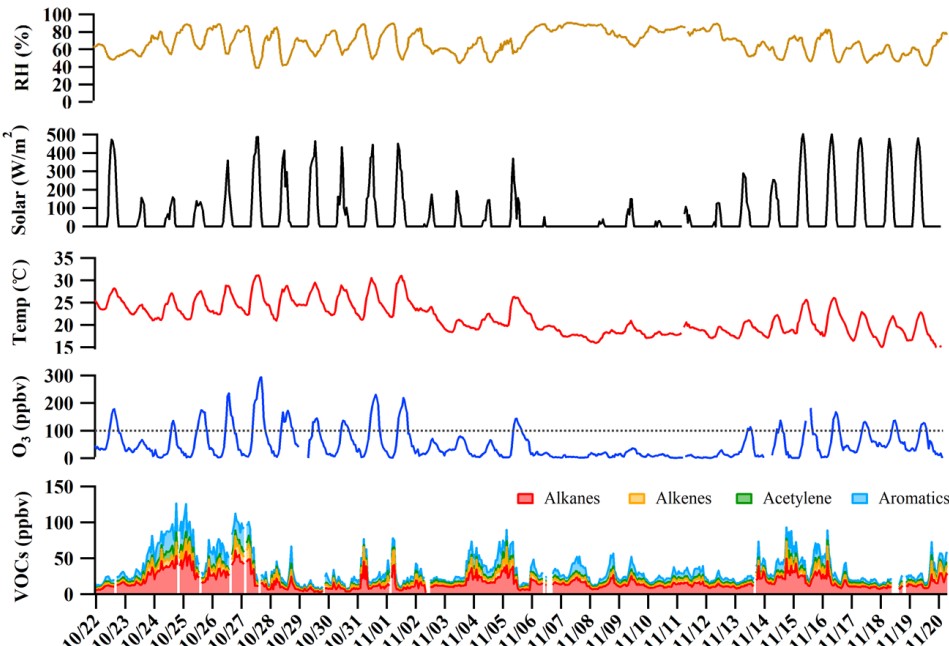


07          Figure 2. Time series of $O_3$, VOCs, and meteorological parameters observed at Heshan site




Table 1. Average, range, and standard deviation of concentrations for the eight most abundant VOCs measured at the

Heshan site, together with a sum of the mixing ratios for each hydrocarbon category (i.e., alkanes, aromatics, and alkenes)

| Species | Average ± standard deviation (ppbv) | Range (ppbv) |
|---|---|---|
| Ethane | 3.86 ± 1.34 | 1.08 - 10.44 |
| Toluene | 3.74 ± 2.89 | 0.56 - 15.80 |
| acetylene | 3.42 ± 2.33 | 0.11 - 28.22 |
| Propane | 3.01 ± 1.82 | 0.45 - 10.70 |
| Ethene | 2.94 ± 3.34 | 0.37 - 64.56 |
| $i$-Pentane | 1.90 ± 2.20 | 0.22 - 16.16 |
| $n$-Butane | 1.85 ± 1.28 | 0.20 - 10.10 |
| $m/p$-Xylene | 1.62 ± 1.44 | 0.17 - 12.74 |
| Alkanes | 17.04 ± 10.64 | 2.04 - 61.01 |
| Aromatics | 9.07 ± 7.01 | 1.46 - 40.12 |
| Alkenes | 5.29 ± 5.01 | 0.67 - 77.39 |

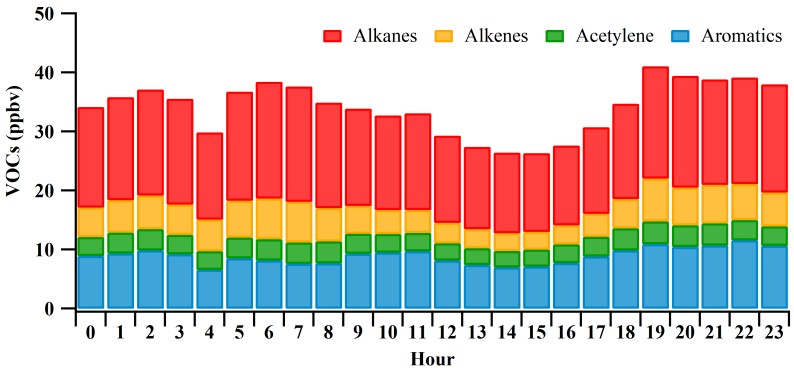

Figure 3. Diurnal variations of VOCs observed at Heshan site

## 3.2 Sources of VOCs

### 3.2.1 The influences of photochemical processing on VOCs concentration

To investigate the source attributions of VOCs, the PMF model was applied to the observed

concentrations of VOCs at the Heshan site. As mentioned above, to more accurately identify and quantify





the source contributions, it is necessary to evaluate whether the photochemical processing could influence

the source signatures of VOCs.

Correlation analysis between two VOC species from the same emissions with different photochemical

reactivity, *i.e.,* propane vs butanes and ethylbenzene vs xylenes, has been widely used in previous studies

to evaluate the photochemistry impacts (Ling et al., 2011). If the two species are involved in

photochemical reactions, their correlation will weaken due to their different reactivity. In the present study,

excellent correlation was found for ethylbenzene vs *o*-xylene ($R^2 = 0.95$, $p < 0.01$) and propane vs *i*-

butane ($R^2 = 0.91$, $p < 0.01$) (figure not shown), indicating the insignificant influence of photochemical

processing on the correlations and clear source signatures for the selected species.

To evaluate further the influence of photochemical processing on the observed levels, a photochemical-

aged-based parameterization method was used to estimate the initial concentrations of VOCs after

emissions (Eq. (5)). This method was first introduced by de Gouw et al. (2005) and was applied to VOC

measured data in different environments (Liu et al., 2009; Shao et al., 2009; Yuan et al., 2012b).

Through the photochemical-aged-based parameterization method, photochemical age, representing the

photochemical processing time, could be calculated by the ratio between the concentrations of two VOCs

with relatively strong correlation and different OH reaction rates, *i.e.,* the ratio of ethylbenzene and *m/p*-

xylene. In this study, high correlation was found between ethylbenzene and *m/p*-xylene ($R^2 = 0.96$, $p <$

$0.01$). Furthermore, the OH reaction rate constants for the above species were $7.10 \times 10^{-12}$ (ethylbenzene),

$1.90 \times 10^{-11}$ (*m/p*-xylene, obtained from the average OH reaction rate constants of *m*- and *p*-xylene)

$cm^3 \cdot molecule^{-1} \cdot s^{-1}$, respectively. It has been demonstrated that ratios of above species could be used to

estimate the effect of photochemical processing on VOC variations (Shiu et al., 2007; Shao et al., 2009;

Chang et al., 2010). The OH exposure ($[OH]\triangle t$) is calculated and used to represent photochemical age,

as $[OH]$ and $\triangle t$ always appear together in the parameterization equation (Jimenez et al., 2009). The OH

exposure is calculated from the ratio of VOCs concentrations by




$$[OH]\Delta t = \frac{1}{(k_E - k_X)} \times \left[ ln \frac{[E]}{[X]} \Big|_{t=0} - ln \frac{[E]}{[X]} \right] \qquad (5)$$
The $[OH]$ term represents the concentration of the OH radical and its reaction time $\triangle t$ for VOCs between
the emission sources and the observation site. The parameters $k_E$ and $k_X$ are the reaction rate constants
of ethylbenzene and *m/p*-xylene, $7.10 \times 10^{-12}$ and $1.90 \times 10^{-11}$ cm$^3$·molecule$^{-1}$·s$^{-1}$, respectively (Atkinson et
al., 2006). $\frac{[E]}{[X]}$ is the average measured concentration ratio of ethylbenzene to *m/p*-xylene. $\frac{[E]}{[X]}\Big|_{t=0}$ is the
initial concentration ratio of ethylbenzene to *m/p*-xylene.
In the present study, the initial concentration ratio of ethylbenzene to *m/p*-xylene ($\frac{[E]}{[X]}\Big|_{t=0}$) is calculated
to be 0.62 using the methods suggested by Yuan et al. (2012b) and Shao et al. (2009), consistent with
those calculated at other urban and rural environments (Shao et al., 2009), and the OH exposure ($[OH]\Delta t$)
calculated by Eq. (5) was $6.47 \times 10^9$ molecule·cm$^{-3}$·s.
On the other hand, the initial concentration of VOCs could be described by
$$[VOC]_{initial} = [VOC]_{measured} \times \exp(-k_{NMHC} \cdot [OH]\Delta t) \qquad (6)$$
Here, *[VOC]initial* and *[VOC]measured* are the initial and measured concentration of particular VOC,
respectively. *kVOC* is the reaction rate constant of the specific VOC.

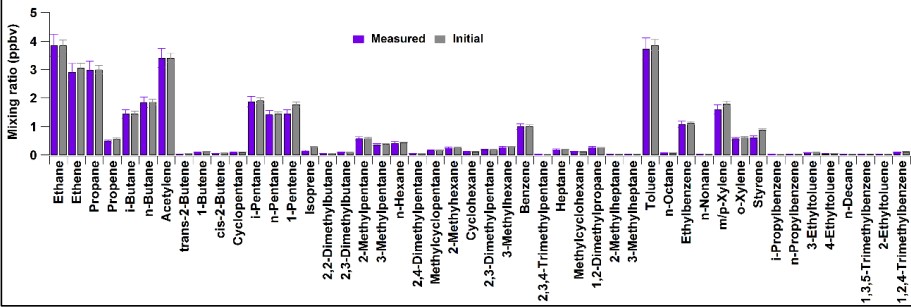


Figure 4. Average measured and initial concentrations of VOCs





Figure 4 shows the comparison between the observed levels and the initial concentrations of VOCs at the

Heshan site. In general, the variations between the observed levels and the initial concentrations of VOCs

were small for most of the VOC species, with the OH reaction rate $< 5.64 \times 10^{-11}$ cm$^3$·molecule$^{-1}$·s$^{-1}$, and

the ratio of initial/observed concentrations ranging from 1.00-1.23. However, for those species with

relatively higher photochemical reactivity (with the OH reaction rate ranging from $5.64 \times 10^{-11}$-$6.40 \times 10^{-11}$

cm$^3$·molecule$^{-1}$·s$^{-1}$), the initial concentrations were 1.44-1.51 times of the observed levels. It should be

noted that these relatively higher reactive species only accounted for a small fraction of the concentrations

and the ozone formation potential (OFP) of all the observed VOCs due to their relatively lower abundance

(data not shown).

Therefore, to consider the influence of photochemical processing on source apportionment results, the

concentrations for the species with relatively higher reactivity (the ratio of initial/observed concentrations >

1.3, *i.e., trans*-2-butene, *cis*-2-butene, styrene, and 1,3,5-trimethylbenzene) were compensated by the

difference between observed levels and initial concentrations. They were further used as input to the PMF

model, together with the observed concentrations of rest species to investigate the source attributions of

VOCs at the Heshan site in the following section.

### 3.2.2 Source apportionments of VOCs

In this study, the data matrix for PMF model was composed of 682 samples and 47 VOCs together with

ACN and MTBE. The solution of four factors was obtained. Figure 5 presents the source profiles (in

percentages of species total) extracted from the PMF model, while Figure 6 presents the relative

contributions of different sources to ambient VOCs at the Heshan site. It was found that factor 1 and 4

were both associated with high percentages of aromatics. In addition to the solvent usage, aromatics were

mainly related to vehicle emissions in the PRD region (Zhang et al., 2013; Ou et al., 2014). The relatively

higher loadings of C$_2$-C$_4$ alkenes in factor 1 suggest that this source was mainly related to diesel vehicular

emission (Guo et al., 2011a; Ou et al., 2014), which accounts for about $25 \pm 3\%$ (mean $\pm$ 95% intervals)

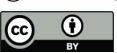



of the total observed VOCs. On the other hand, factor 4 was characterized by high levels of *n/i*-pentane

and MTBE, the typical tracers for gasoline vehicular emissions (Song et al., 2006; Ho et al., 2009; Ou et

al., 2014). As such, factor 4 was assigned to gasoline vehicular emission and its contribution to the

observed VOCs was $33 \pm 5\%$. Factor 2 was characterized by high percentages of $C_6$-$C_7$ alkanes and certain

amounts of aromatics, while the contributions of other combustion tracers were insignificant in this factor,

suggesting that this factor was related to solvent usage. It is consistent with previous studies that aromatics

and $C_6$-$C_7$ could be used as solvents in the printing and paint industry (He et al., 2002, Chan et al., 2006;

Liu et al., 2008). $18 \pm 2\%$ of the observed VOCs, were identified to be associated with solvent use. Factor

3 was represented by high percentages of ethane (~65%), acetylene (~51%), benzene (~52%), and ethene

(~ 31%), together with some $C_3$-$C_5$ alkanes and alkenes, which are typically tracers of incomplete

combustion such as vehicular exhaust and biomass burning (Nelson et al., 1984; Wadden et al., 1986;

Blake et al., 1994; Rudolph, 1995; Guo et al., 2011a, 2011b). The high percentage of acetonitrile in factor

suggests that this factor was associated with biomass burning (Holzinger et al., 1999; Yuan et al., 2010),

which was responsible for $24 \pm 3\%$ of the observed VOCs. Figure 7 shows the diurnal variations of VOCs

emitted from different sources extracted from PMF. Different diurnal patterns were found for different

sources, which may be related to the variations in emission strength, the concentrations of species in

different source profiles, as well as the influence of mixing height. For example, relatively higher levels

were found for the diesel and gasoline vehicular emissions in the early morning and in the evening,

corresponding well with traffic emissions during rush hours, while a broad peak during daytime hours

may be related to the increased mixing height and photochemical loss, and decreased emission strength

(Zheng et al., 2010; Yuan et al., 2009). Different from vehicular emissions, the concentrations of solvent

usage started to increase in the early morning and reached maximum value at noon, and then decreased

gradually and presented a broad peak until midnight. The increased levels of solvent usage from early

morning to midday was associated with the increased emissions from human production activities and the

increased temperature which would accelerate VOCs evaporated during the use of solvent. The diurnal



variations of biomass burning were much weaker compared with other sources, with peak values occurred
in the early morning, which was consistent with the diurnal patterns of plumes of biomass burning at
Heshan site (Yuan et al., 2010).

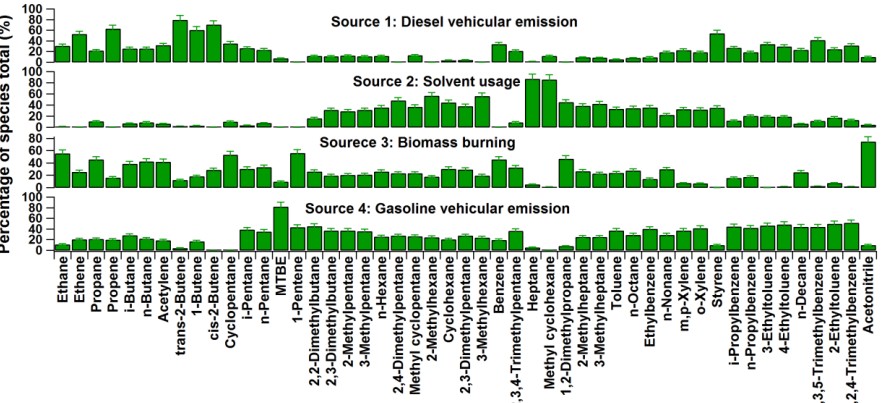

11                         Figure 5. Factor profile (in percentage of species total) attributed from PMF

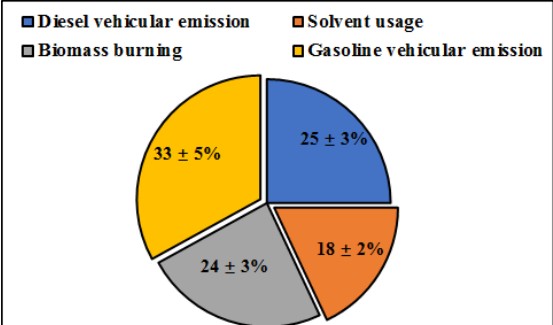

13                         Figure 6. Contributions of different sources to ambient VOCs extracted from PMF

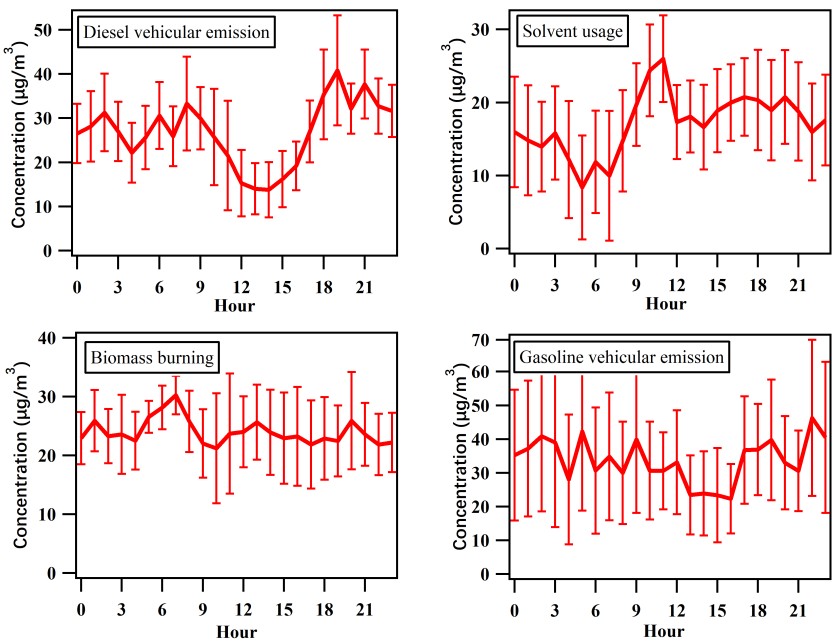

15          Figure 7. Diurnal variations of VOCs emitted from different sources extracted from PMF

**3.3 The contributions of VOCs sources to ozone photochemical formation**
To evaluate the roles of the different VOCs emissions for $O_3$ formation, we applied the PBM-MCM model
to the PMF-extracted VOC concentrations. Figure 8(a) shows the average RIR values of different VOC
sources and NO, together with the contributions of different VOC sources to photochemical $O_3$ formation.
The average RIR values of various VOC sources were positive, while that of NO was negative, suggesting
that $O_3$ formation at Heshan site was located in the VOC-limited regime. Among the four main
anthropogenic sources of VOCs, relative higher average RIR values of vehicular emissions and biomass
burning than that of solvent usage were found. Furthermore, considering both the reactivity and
abundance of VOCs in different sources, the relative contributions of the four anthropogenic sources were





calculated by Eq. (3) and the results are shown in Figure 8(b). It shows that the vehicular emissions,
including diesel and gasoline vehicular emissions, made the most important contributions to
photochemical $O_3$ production, with an average percentage of 58%, followed by biomass burning (28%)
and solvent usage (14%), suggesting that controlling vehicular emissions and biomass burning could be
more effective way for reduction of $O_3$ pollution at Heshan.

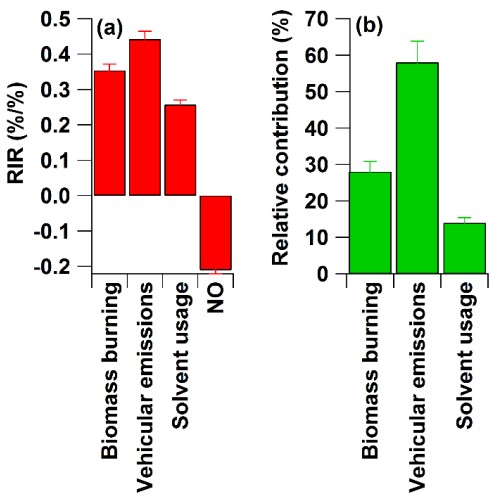

Figure 8. The average RIR values of VOC sources and NO, together with the contributions of different VOC sources to
photochemical $O_3$ formation at Heshan
**3.4 Improvement for the reduction of VOCs and $NO_x$ to photochemical $O_3$ formation**
**3.4.1 Sensitivity analysis of ozone formation**
Changes in the concentrations of VOCs and $NO_x$ will affect $O_3$ formation, leading to a great variation of
the $O_3$ concentration, which can be illustrated from the ozone isopleth plot. The PBM-MCM model was
employed to perform a sensitivity study based on the average diurnal variations of the observed air
pollutants (*i.e.*, 58 VOCs, $O_3$, NO, $NO_2$, CO, and $SO_2$.) at the site. We followed the procedures suggested

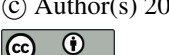



by Lyu et al. (2016) to investigate the $O_3$ variations related to the changes of precursors. A total of 520
reduction scenarios (26 $NO_x$ × 20 VOCs) were simulated and the maximum $O_3$ value in each scenario
was selected. Figures 9 shows the obtained ozone isopleth based on the observed levels of VOCs and $NO_x$
at Heshan site.
The two ozone isopleths (Figures 9(a) and (b)) show distinct characteristics of $O_3$ variations. For a 0-60%
reduction of $NO_x$ (corresponding to 40-100% of original $NO_x$), the $O_3$ mixing ratio decreases significantly
with VOCs for a certain $NO_x$ condition but increases slightly with decreasing $NO_x$ for a fixed mixing ratio
of VOCs (Figure 9(a)). This clearly indicates a VOC-limited regime for $O_3$ formation in this region and
is consistent with previous results found in the urban, suburban, and even some rural environments, as
well as the downwind site of the PRD region (Zhang et al., 2008b; Cheng et al., 2010; Ling et al., 2011;
Zheng et al., 2013; Ling and Guo, 2014). However, it is different from the results found in the northern
rural areas of the PRD region based on the measured ratios of $O_3/NO_x$ (Zheng et al., 2010). Here, we
introduce the absolute value of RIR (|RIR|) to evaluate the sensitivity of the $O_3$ formation to VOCs and
$NO_x$. It turns out that the |RIR| decreases with VOCs for a fixed $NO_x$, while it fluctuates at first and then
increases with decreasing $NO_x$ for a fixed VOCs. This implies that the efficiency of $O_3$ reduction by
cutting down VOC emissions alone would decrease gradually (data not shown) and we may need to pay
attention to the counter effects caused by decreasing $NO_x$. For $NO_x$ reduction to 0-40% of the original
level, a clear ridge can be seen (Figure 9(b)), dividing the isopleth into two parts: VOCs-limited (right)
and $NO_x$-limited (left). In the VOCs-limited regime, the effects of both VOCs and $NO_x$ on $O_3$ formation
are linear, and the ozone concentration is apparently proportional to the amounts of VOCs (and $NO_x$): the
higher the VOCs mixing ratio (or the lower the $NO_x$), the higher the $O_3$ concentration. On the other side
of the ridge, for the reduced $NO_x$ to 7.5-15% of original mixing ratio, $O_3$ concentration decreases with
$NO_x$ concentration, but VOCs decrease would lead to $O_3$ increase. Ideally, the NOx reduction to its 1-
7.5%, a regime with "pure" $NO_x$-limited would occur, where ozone formation is solely controlled by the



NO$_x$ concentrations and insensitive to the change of VOCs. The reduction of NO$_x$ emissions in this case
will be the most effective measure for mitigating ozone production.

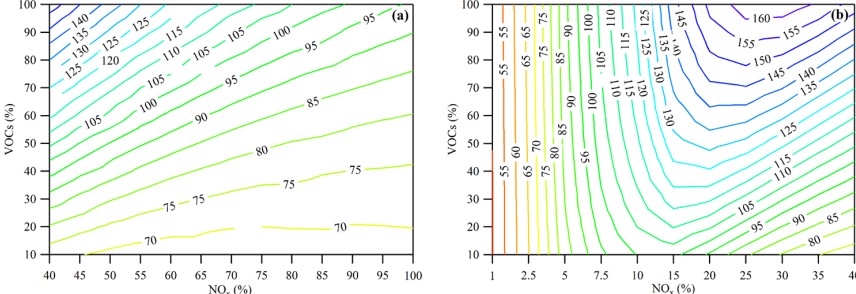


Figure 9. The ozone isopleth in term of percentage changes of VOCs and NOx: the percentage change of NO$_x$ from 0 to
60% (a) and from 60 to 99% (b). The ozone mixing ratios are in ppbv. The horizontal and vertical axes correspond to
the percentage of the measured average mixing ratios of NO$_x$ and VOCs, respectively.
**3.4.2 Development of the most optimum control measures on both VOCs and NO$_x$**
Though it was found that the O$_3$ formation was located in the VOCs-limited regime at the Heshan site
(with 100% of NO$_x$ and VOCs as input), it is unknown how much VOCs should be controlled for the
most efficient O$_3$ reduction, especially in society where VOC$_s$ and NO$_x$ frequently are controlled
simultaneously. To achieve this goal and provide detailed information about the amounts of VOCs and
NO$_x$ that need to be controlled, we simulated the net O$_3$ increment (the total increase of average O$_3$
concentrations when both VOCs and NO$_x$ are reduced) with the reduction of both VOCs and NO$_x$. This
is shown in Figure 10. The horizontal and vertical axis corresponds, respectively, to the reduction
percentages of NO$_x$ (e.g., 10% means that the mixing ratios of NO$_x$ were reduced by 10%) and the net
increments of O$_3$ (positive and negative values represent the increase and decrease of O$_3$ compared to the
base case with no reduction of VOCs and NO$_x$, respectively). The different curves correspond to scenarios
with different cutting percentages of VOCs. It shows that the net O$_3$ increments increased as the reduction
percentages of NO$_x$ increased from 0 to 70% regardless of the reduction of VOCs, while the net O$_3$




increment decreased gradually, starting from the $NO_x$ reduction percentage of ~70%. However, an
optimum control measure for VOCs and $NO_x$ was that when this control measure was conducted, the $O_3$
mixing ratios would be reduced or at least the $O_3$ mixing ratios would not increase (*i.e.,* the value of the
net $O_3$ increment was less than or equal to zero, the highlighted area in Figure 10). It was found that when
the mixing ratios of VOCs were reduced from 0 to 99.5%, the appropriate reduction percentages of $NO_x$
should be located between 0 and 88% or between 90 and 99.5% for zero $O_3$ increment. However, it was
interesting to find that when the reduction percentages of $NO_x$ ranged from 90 to 99.5%, the $O_3$ formation
reduced with the reduction of $NO_x$ regardless of the reduction of VOCs, though reducing $NO_x$ by 90~99.5%
is probably unrealistic. Therefore, this section only focused on the range of 0~88% of $NO_x$ reduction for
devising optimum controlling measures of VOCs and $NO_x$. It was determined that when the reduction
percentages of $NO_x$ ranged from 0 to 88%, the minimum abatement ratio of VOCs/$NO_x$ for zero $O_3$
increment changed from ~1 to 1.1 (i.e., the cutting ratios of VOCs/$NO_x$ at the intersections of the curves
and the horizontal axis). This suggests that the abatement ratio of VOCs/$NO_x$ should be more than 1.1 to
prevent the increase of the $O_3$ levels at Heshan.

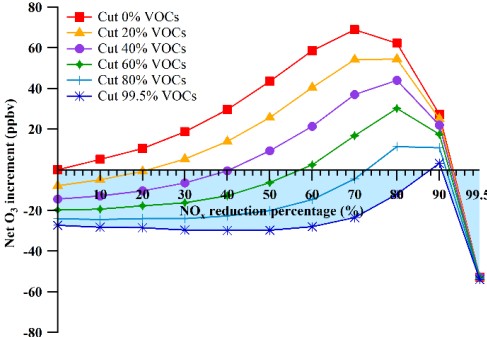

Figure 10. Net $O_3$ increment as function of the reduction percentages of $NO_x$ and VOCs. The highlighted area represents
zero $O_3$ increment

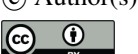



Furthermore, to determine the appropriate cutting ratios of the individual source of VOCs vs $NO_x$ when both VOC sources and $NO_x$ were reduced, $O_3$-VOC sources-$NO_x$ sensitivity analysis was conducted. The net $O_3$ increments as a function of the reduction percentages of $NO_x$ for the individual VOC source are shown in Fig. 11. For different VOC sources, the patterns of the net $O_3$ increment as a function of different $NO_x$ reductions are similar, with $O_3$ increments increased when the mixing ratios of $NO_x$ was reduced by 0~80% and decreased with the reduction percentages of $NO_x$ ranging from 80 to 99.5%. It was found that when the cutting percentages of VOCs increased from 0 to 99.5%, the appropriate reduction percentages of $NO_x$ for zero $O_3$ increment were located in the ranges of 0~30% and 90~99.5%. It was interesting to find that when the reduction percentages of $NO_x$ ranged from 90 to 99.5%, the $O_3$ formation reduced with the reduction of $NO_x$ regardless of the reduction of VOCs, though reducing $NO_x$ by 90~99.5% is probably unrealistic in the near future. Therefore, here we only focus on the range of 0~30% of $NO_x$ reduction to provide appropriate reduction ratios for devising optimum controlling measures of VOCs and $NO_x$.

It was found that the specific appropriate reduction percentages of $NO_x$ for zero $O_3$ increment was 0~27%, 0~22%, 0~22%, and 0~30% for diesel vehicle emission, solvent usage, biomass burning, and gasoline vehicular emission, respectively, while the abatement ratio of individual sources of VOC vs $NO_x$ should be more than 3.8, 4.6, 4.6, and 3.3 for diesel vehicular emission, solvent usage, biomass burning, and gasoline vehicular emission, respectively. For example, if the mixing ratios of $NO_x$ were reduced by 10%, more than 38% of diesel vehicular emission, 46% of solvent usage or biomass burning, or 33% of gasoline vehicular emission needs to be cut to prevent increase of $O_3$ levels at Heshan. Furthermore, the above ratios demonstrate that reducing VOCs from gasoline vehicular emission has the highest efficiency for both reductions of VOCs and $NO_x$ without increasing $O_3$ levels, followed by diesel vehicular emission, biomass burning, and solvent usage.





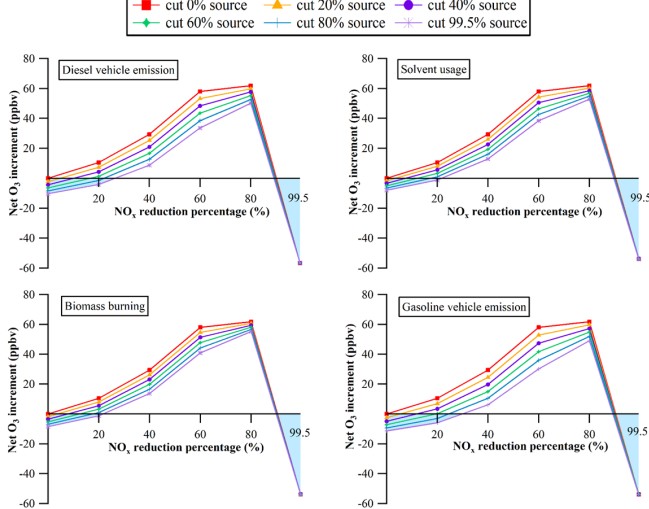

Figure 11. Net O₃ increment as a function of the reduction percentages of NOₓ for individual sources of
VOC. The highlighted area represents zero O₃ increment
**4 Conclusion**
The PRD region has long been facing severe photochemical air pollution, and VOCs has been the limiting
factor of ozone formation in this region. To better understand the contribution of different anthropogenic
VOCs to the ozone formation in this region, we performed in-depth analysis on the data of intensive
measurement of VOCs and related species conducted at a downwind rural site (the HeShan site) of the
PRD region during October to November of 2014. Overall the mixing ratio of total measured VOCs was
high and similar to other urban cluster regions. By using the PMF model with consideration of
photochemical processing effects, four anthropogenic emissions sources were identified. The vehicular
emission was the most important source of VOCs and also the major contribution to O₃ formation at
Heshan site, followed by biomass burning. The PBM-MCM model analysis confirms that the ozone
formation at Heshan was VOC-limited, and the regime would not transform until the NOₓ is reduced to ≤




25% of the current concentration. The $O_3$-VOCs-$NO_x$ sensitivity analysis in the whole air suggested that >1.1 of the abatement ratio of VOCs/$NO_x$ was the most appropriate abatement ratio when both the VOC and $NO_x$ were reduced to prevent net $O_3$ increment. Furthermore, $O_3$-VOC sources-$NO_x$ sensitivity analysis suggested that the cutting ratios of individual source of VOC vs $NO_x$ should be more than 3.8, 4.6, 4.6, and 3.3 for diesel vehicular emission, solvent usage, biomass burning, and gasoline vehicular emission, respectively, for the more effective control of the $O_3$ formation, which providing important scientific support for the management department for better formulation and implementation of measures on photochemical pollution.

Indeed, more and more policies on VOCs were/are being implemented and formulated in the PRD region and/or the whole country. A series of policies regarding the control of vehicular emissions have been conducted in the PRD region, which can be mainly divided into two categories: 1) Improve the environment standards of the main air pollutants, *e.g.*, the National Ambient Air Quality Standard of GB 3095-2012; 2) Improve the quality of the fuel used in vehicles, *e.g.* the fifth phase of the vehicle emission standards (GB 18176-2016, GB 14622-2016, GB 19755-2016, and HJ 689-2014). It should be noted that the fifth phase of the vehicle emission standards have limited the emissions of total VOCs from vehicles (IGES, 2014) compared with those of the fourth phase. On 26 December 2017, the Monitoring Plan on the Ambient Volatile Organic Compounds (VOCs) in Key Areas in 2018 was issued by the Ministry of Environmental Protection, which will help to better supervise and provide more observed data for exploring the effective pathways to alleviate VOCs and photochemical $O_3$ pollution. In addition, "new energy automobiles" are promoted widely in urban cities of the PRD region, such as Guangzhou and Shenzhen, and the eight civil conduct codes of "Breath and Struggle together" were released to raise the public's concern about how to improve the air quality. However, the policies on controlling biomass burning are relatively limited when compared to those for vehicular emissions. For biomass burning, straw return and collected integrated agricultural machinery have been encouraged and widely promoted, and the outdoor burning of straw was forbidden according to the Law of the Prevention and Control of



Atmospheric Pollution. The emission standards for biomass burning are included in the latest version of
"Air pollution emission standard for boilers" (GB 13271-2014), where emission standards of coal-fired
boilers were applied to biomass briquettes. Furthermore, the emissions from biomass forming fuel boilers
and the biomass-molded fuel used in Guangdong should meet DB44/765-2017 and DB44/T 1052-2012,
respectively.
Nevertheless, it is expected that these measures could help alleviate the photochemical pollution and
improve air quality and visibility, but most of them only control the total mass of VOCs. Therefore,
evaluation on the benefits of these measures on VOCs and photochemical pollution is still needed. Overall,
the findings of this study provide quantitative information on devising appropriate measures on the VOCs,
$NO_x$, and $O_3$ pollution at the receptor site of PRD, which could be extended to other regions in China.

**Competing interests**

The authors declare that they have no conflict of interest.

**Author Contributions**

**I**n this study, the analysis methods were developed and the whole structure for the manuscript was
designed by Dr. Zhenhao Ling, Dr. Zhe Wang and Prof. Jun Zhao. Ms. Zhuoran He conducted the data
process and wrote the original copy of the manuscript. Prof. Xuemei Wang and Prof. Min Shao provided
the data and revised the manuscript. Furthermore, the simulation of PBM-MCM model was conducted by
Dr. Zhenhnao Ling and Prof. Hai Guo. Finally, the manuscript was finalized by Dr. Zhenhao Ling and
Dr. Zhe Wang.



## Acknowledgements

This study was supported by the National Key Research and Development Program of China (2017YFC0210106, 2016YFC0203305), the State Key Program of National Natural Science Foundation of China (No. 91644215), the National Natural Science Foundation of China (No. 41775114 and 41505103) and Hong Kong Research Grants Council (25221215, 15265516).

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
