# Peer review of "Contributions of different anthropogenic volatile organic compound sources to ozone formation at a receptor site in the Pearl River Delta region and its policy implications"

_Atmospheric Chemistry and Physics, 2018_

## Referee Comment (RC1) · Anonymous Referee #3 · 8 Mar 2019

General comments

The manuscript enhances current understanding of ozone formation in the Pearl River Delta, China. The authors analyze one month of measurements made at a receptor site using positive matrix factorization (PMF) and a parameterization of photochemical age to identify and quantify contributions of four emission source types to VOC mass concentration. A box model is used to calculate relative incremental reactivities (RIR) and source contributions to ozone formation. The model is also used to develop ozone response surfaces as a function of VOC and NOx emission reductions.

[Figure]

The manuscript states that one important motivation for their study is the need to determine how photochemical processing affects the accuracy of PMF applications to VOC source apportionment in the study region (p 4, l 80 - 85). It would be informative to know how much difference the calculation of photochemical processing made to the PMF results (sections 3.2.1 and 3.2.2). A simple comparison of PMF results with and without the adjustment for photochemical losses would clarify how important it is to do the photochemical loss calculation and adjustment of species concentrations. Such a comparison might make the manuscript more general and of greater interest to a wider audience.

The authors correctly note that their "results could provide valuable information, facilitating local and regional policy-makers. . ." Their findings are indeed valuable for the specific region studied. The manuscript could have wider significance if the authors provided more comment about how their methods or their results would be of interest to researchers or policymakers in other geographical areas.

The absence of biogenic VOC (e.g., isoprene) contributions to ozone formation is of concern (please see specific comments). Since the results were determined for a one-month autumn sampling period, additional discussion of the applicability of the findings to other seasons is merited.

Readers will need some additional information to evaluate the accuracy of the box model simulations (please see specific comments).

Specific comments

Please note that the "hundreds" digits of the line numbers are not visible in the PDF, so I refer to both page number and line number throughout this review.

p. 7, l 32 – 34. The exclusion of biogenic VOC measurements as inputs to PMF is a limitation. Because the measurement period was October 22 to November 20, 2014, it is possible that biogenic emissions were much lower than would be typical of spring

or summer and the exclusion of biogenic VOC concentrations from the PMF inputs may not affect the conclusions for this study period. However, there should be some acknowledgment that the results pertain to a fall study and may not represent other seasons.

p 8, l 52 – 54. Please clarify if biogenic VOCs were included among the inputs to the box model. The RIR values will not be correct if biogenic VOCs were excluded from the inputs to the box model.

p 8, l 52 – 54. Please clarify that the model was applied to each individual day in the study for the RIR calculations (rather than that hourly data were averaged across days and then input as averages to the model as was later done to generate the ozone isopleths). The citations to Lyu et al. (2016) and Wang et al. (2017) imply that the model was applied to each individual day (as done in those studies) for the RIR calculations but it would be helpful to be explicit in this paragraph.

p 8, l 64. It might be helpful to say "net O3 production plus NO consumed" rather than "net O3 production and NO consumed" (the equations are correct).

P 9, l 95 – 96. If O3 formation at the site is VOC-limited as previously discussed, it isn't likely that photochemical processing of the arriving air masses is truly complete. Even if NOx mixing ratios are too low to sustain further O3 production in the arriving air, injection of fresh NOx emissions into aged air masses would permit further photochemical processing. The next section (3.2.1) indicates that photochemical aging was relatively incomplete (high correlations between more reactive and less reactive species; similarities between initial and observed VOC species concentrations except for highly reactive species such as some alkenes and aromatics).

P 11, Table 1. It would be helpful to include summary statistics for NO and NO2.

P 17, section 3.3. For readers to understand the accuracy of the box model, it would be helpful to provide model performance statistics for the base case at the beginning

of this section.

P 18, Figure 8 and related text. The RIR calculation for each species depends on the concentrations of all other species. If biogenic VOCs (e.g., isoprene) were not included as inputs to the box model, the reported RIR values will not represent the "real-world" condition. If biogenic VOCs were included as inputs to the box model, Figure 8 should show their RIRs and and relative contributions. Please note in the caption what the error bars represent (e.g., one or two standard errors of the means or a different measure of variation?)

P 18, l 37 - 38. The phrase "based on the average diurnal variation" suggests that hourly data were averaged across days to provide a single base-case input for the box model's generation of ozone isopleths, which would be consistent with the approach described by Lyu et al. (2016). Please expand this description slightly. Do the 58 VOCs include isoprene and alpha- or beta-pinene?

P 19, l 61. Even in the NOx-limited regime, VOC reductions never increase ozone formation. Consistent with past studies, the isopleths are nearly vertical on the left side of Figure 9b and indicate a very small ozone decrease in response to VOC reductions at fixed NOx.

Pages 21 – 22. Many readers will not be able to follow this discussion as written.

p. 23, l 34 – 35. Does this conclusion apply to other seasons?

Minor comments

Abstract. The abstract uses a term ("abatement ratio") that is defined in the text (page 21), requiring a full page of explanation there. Another full page (p 22) is needed to define and discuss the abatement ratios of the individual source types. To aide readers of the abstract who will not have already read pages 21 - 22, I suggest revising the following sentence:

"Sensitivity analysis indicated that in order to prevent the increment of O3 concentration, the abatement ratios of the individual VOC source vs. NOx should be higher than 3.8, 4.6, 4.6, and 3.3, respectively, for diesel vehicular emission, solvent usage, biomass burning, and gasoline vehicular emission, respectively."

A sentence such as the following appears to me to better convey the intended meaning: "Sensitivity analysis indicated that combined VOC and NOx emission controls would effectively reduce incremental O3 formation when the ratios of VOC-to-NOx emission reductions were higher than 3.8 for diesel vehicular emission, 4.6 for solvent usage, 4.6 for biomass burning, and 3.3 for gasoline vehicular emission."

P 3, l 71-73. Please clarify if the percentages refer to percent of VOC mass or percent of VOC reactivity.

P 5, l 99-100. The map shows that the site is southwest of Guangzhou and Foshan, not northeast (alternatively, the cities are northeast of the site).

P 13, eq. 6 and l 53 – 54. For consistency, please use either kVOC or kNMHC in both eq. 6 and the text.

p. 20. Figure 9 is confusing for two reasons. First, it would be easier to understand the figure if the two panels were swapped (right replacing left). Second, the axes are also reversed from the more common presentation format. Reversing the vertical and horizontal axes and then placing the current left panel above the current right panel would be consistent with customary presentations (NOx reductions on the vertical, VOC reductions on the horizontal axis).

---

## Referee Comment (RC2) · Anonymous Referee #1 · 8 Mar 2019

The authors presented hourly resolved measurements of volatile organic compounds (VOCs) at a receptor site in the Pearl River Delta (PRD) region in China. A receptor model, positive matrix factorization (PMF), was used to apportion sources of VOCs, by taking into account of photochemical degradation of some VOCs. Results showed that four sources, including gasoline vehicular emission, diesel vehicular emission, biomass burning, and solvent usage, are major contributors to anthropogenic VOCs at the site. A photochemical box model with the master chemical mechanism was also used to evaluate the contributions of those VOCs to ozone formation, with vehicular emission

found to be the most significant. Further analysis to test abatement scenarios was also performed for policy implication.

Overall, this is a nice data set with thorough analysis. The manuscript is generally well written and is surely of interest to readers of ACP. I therefore recommend minor revision before publication, with below comments for the authors.

Major:

1. I do not understand that in Section 3.3, why did the authors sum up gasoline and diesel vehicular emissions in their discussion on the contributions of different VOC sources to ozone formation. It is of importance for policy making, as to which type of vehicular to control (e.g., with priority), if we have more detailed understanding on whether vehicles run on gasoline or diesel have more potential in VOC emission that is related to ozone formation. Can the authors justify and clarify?

2. Using the same set of data, the same group of authors published in Journal of Environmental Science (JES) recently. Although the foci of the two papers are different, with the JES paper on isoprene and their oxidation products and this one on anthropogenic VOCs, I do like to see some connection between the two papers as they are based on the same data set (it was not even cited here). More importantly, what is the similarity and difference in methodology between these two papers? Would there be any bias if the whole chunk of biogenic VOCs were taken out from photochemical box model (e.g., the source/fate of OH radicals and ozone)?

3. Page 12, line 19-25. I do not understand the assertion here that correlation will be distorted if two VOCs from the same source have different photochemical reaction rates. The authors used the photochemical age concept in the paragraph right after it, which means that correction can still be retained if the two VOCs react in a proportional manner with OH radicals (and assuming no other fates). The rationale of these two paragraphs seems contradicting. Please clarify.

4. P7/L37: I would strongly suggest the authors include more details on why a four-factor solution was chosen. Diagnostic analysis by comparing three-factor and five-factor solutions would be useful, even as supplementary materials. "a good fit to the data and the most meaningful results" is just too descriptive and not very convincing.

Minor:

1. P2/L41: "complex, nonlinear" to "complex and nonlinear".

2. P2/L44: "VOCs and NOx limited" to "VOC- and NOx-limited"; "VOCs-limited" to "VOC-limited", and in other places as well.

3. P3/L60: "emission-inventory" to "emission inventory".

4. P4/L84: did Ling et al., JES, 2019 take photochemical processing into account?

5. P6/L19: remove "," after "contributions".

6. P7/L32: "the detection limit" to "their detection limits".

7. P7/L35: why MTBE and ACN so special and not considered as "VOCs"?

8. P8/L58: add "divided by" after "production"?

9. P9/L89&L91: "that" to "those".

10. P9/L94-96: how can variations of VOCs suggest photochemical processing? It could be just variations on sources. Please clarify.

11. P9/L96: delete "to be".

12. P12/L34-35: these rate constants appeared later in the next page and look redundant. Please remove.

13. P13/Eq 6: k_VOC instead of k_NMHC? It would be good to have a table showing the rate constants for each VOCs and appropriate citation.

14. P14/L60&L62: "reaction rate" should be "reaction rate constant"?

15. P14/L71: add "the" before "rest species".

16. P15/L89: remove "," after "VOCs".

17. P15/L93: "acetonitrile" to "ACN" (you defined it early).

18. P15/L00: "peak" or "valley"?

19. P17/L22: "relative" to "relatively".

20. P18/L29: add "a" before "more".

21. P23/L31: remove "cluster".

22. P24/L44-P25/L65: I would suggest to shorten this paragraph to a few sentences to make the point: although there are some control measures on VOC emission from vehicles, there is limited control on biomass burning etc.

23. Figure 4: can the authors use another panel to show ratios too?

24. Figures in general: it would be more reader friendly if the authors can use a bigger font size for most of the figures.

---

## Referee Comment (RC3) · Anonymous Referee #2 · 11 Mar 2019

This manuscript presents an analysis of hourly VOC data from the Pearl River Delta region including PMF and VOC sources important for ozone formation. There is also an analysis of VOC and NOx limitation and what effect controlling each of these may have on ozone. In general the manuscript is well written with only a few sections needing clarification.

Specific Comments: P8 Line 62-63: Is this a result from this study or from elsewhere? It is confusing to have these papers cited without more explanation.

[Figure]

Table 1: Add a line dividing the individual species from the categories. This will make the table easier to read.

P9 Line 73-78: The use of these correlations seems to be overstated both in terms of widespread use (1 paper cited) and what these correlations mean for photochemistry impacts. Weak correlations may exist due to strength of impact from different sources in addition to changing photochemistry. Adding some statements about the degree of correlation or lack of correlation between other species would strengthen this section. A more convincing argument would be to look at photochemically formed species from these precursor compounds – I'm not sure if these compounds are available.

This simple analysis and the extended analysis using the parameterization suggests little photochemistry is taking place. Is this due to the time available for photochemistry to take place or atmospheric conditions. What is the estimated air mass age? (using chemical tracers, proximity to sources and average wind speed, or from the parameterization).

Figure 2: Consider showing as the difference between the measured and estimated initial concentration instead. It will be easier to see the delta for all species than infer it from the difference between bars.

Figure 5: Why are both diesel and gasoline vehicles together if they were different factors?

Sect 3.3 How does this compare for the different classes of VOCs? This would be good to include to inform which measurements would be most important to make long-term or in the future. Or if new industry moved into the area and the mix of VOCs was different.

Sect 3.4.2 – something about the tenses used (past and present) is confusing or not consistent. I suggest going through the section and checking verb use for consistency and clarity.

Line 19 Line 40: Which of your reduction scenarios reflect these changes? I ask that for the majority of this section. The listing of all these policies without more direct connection to your findings is superfluous. Only the last paragraph in the section touches on this but still doesn't connect how much these policies are expected to change VOC and NOx levels. The importance and relevance of this section needs to be considered. If it is important figure out a way to make it easier to follow and more connected to the rest of the paper.

P9 Line 77: little instead of insignificant. Insignificant implies statistics were used.

P11 Line 11: adjusted instead of compensated

P12 Line 26: C6-C7 alkanes?

P13 Line 39: why "again"? I don't think you've presented PBM-MCM data yet.

P19: Line 32-33: This suggests. . .in the region.

P19 Line 42: closely instead of strictly

P19Line 42: What are new energy automobiles?

―――――――――――――――――

---

## Author Comment (AC1) · 20 May 2019

Response to Reviewers We appreciate the two anonymous reviewers for their constructive criticisms and valuable comments, which were of great help in improving the quality of the manuscript. We have revised the manuscript accordingly and our detailed responses are shown below. All the revision is highlighted in the revised manuscript.

Reviewer #3

General comments The manuscript enhances current understanding of ozone forma-

tion in the Pearl River Delta, China. The authors analyze one month of measurements made at a receptor site using positive matrix factorization (PMF) and a parameterization of photochemical age to identify and quantify contributions of four emission source types to VOC mass concentration. A box model is used to calculate relative incremental reactivities (RIR) and source contributions to ozone formation. The model is also used to develop ozone response surfaces as a function of VOC and NOx emission reductions.

The manuscript states that one important motivation for their study is the need to determine how photochemical processing affects the accuracy of PMF applications to VOC source apportionment in the study region (p 4, | 80 - 85). It would be informative to know how much difference the calculation of photochemical processing made to the PMF results (sections 3.2.1 and 3.2.2). A simple comparison of PMF results with and without the adjustment for photochemical losses would clarify how important it is to do the photochemical loss calculation and adjustment of species concentrations. Such a comparison might make the manuscript more general and of greater interest to a wider audience. Reply: The reviewer's suggestion is highly appreciated. Figure 4 was revised in order to present the difference on VOCs concentrations with and without the adjustment. Furthermore, in order to investigate the influence of photochemical processing on the source apportionment of VOCs, a comparison of PMF results with and without the adjustment of VOC concentration due to photochemical losses was conducted. Similar sources profiles and the same four anthropogenic sources (i.e., diesel vehicular emission, solvent usage, biomass burning, and gasoline vehicular emission were identified, however the sources made different contributions to VOC abundance compared to those with the adjustment of VOC concentrations (more details were provided in the supplementary). For example, higher contributions of solvent usage and biomass burning and lower contributions of diesel vehicular emission were found in the scenario without adjustments than that with adjustments, which were related to the relatively higher photochemical reactivity of main species in diesel vehicular emission than those in solvent usage and biomass burning. The above brief discussion

on the difference in source apportionment results with and without the adjustment of VOCs has provided in the revised manuscript (Lines 4-13, Page 17), while the detailed discussion was provided in the supplementary.

The authors correctly note that their "results could provide valuable information, facilitating local and regional policy-makers..." Their findings are indeed valuable for the specific region studied. The manuscript could have wider significance if the authors provided more comment about how their methods or their results would be of interest to researchers or policymakers in other geographical areas. Reply: The reviewer's suggestion is highly appreciated. To highlight the significance of this study and provide more statement on how the results of this study would be of interest to researcher or policymakers in other geographical areas, the conclusion section was combined with the policy implication section as follows: "The PRD region has long been facing severe photochemical air pollution, and VOCs has been the limiting factor of O3 formation in this region. To better understand the contribution of different anthropogenic VOCs to O3 formation in this region, we performed in-depth analyses on data obtained from intensive measurements of VOCs and related species conducted at a downwind rural site (Heshan site, HS) of the PRD region during October – November, 2014. Four anthropogenic sources were identified by the PMF model with the consideration of the influence of photochemical processing. The O3 formation at the HS was generally VOC-limited, with the vehicular emission (especially gasoline vehicular emission) as the most important anthropogenic VOC source contributing to O3 formation, followed by biomass burning. It indicated that priority should be given to controlling vehicular emission and biomass burning. Furthermore, with the current industries operating in the PRD region, particular attention should be given to toluene, xylenes, ethylbenzene, ethene and 1-pentene in efforts to control photochemical pollution. Indeed, many additional policies on VOCs have been and continue to be implemented and formulated in the PRD region. A series of policies regarding the control of vehicular emission have been conducted in the PRD region, the purposes of which can be mainly divided into two categories: 1) improve the environmental standards of the main air pollutants

and standards of emissions; and 2) improve the quality of the fuel used in vehicles. Policies on controlling biomass burning, however, are relatively limited. Nevertheless, some policies have been effective, and levels of NOx (another important O3 precursor), have decreased in the PRD region in recent years. On the other hand, for VOCs, most relevant policies only control the total mass and/or the total emissions of VOCs, and the level of O3 continues to increase in this region. To prevent net O3 increment, the VOCs and NOx should be controlled in an appropriate ratio since VOCs and NOx were frequently controlled simultaneously. Furthermore, long-term monitoring is still needed to evaluate the benefits and dis-benefits of the control measures on vehicular emissions and/or photochemical pollution in the PRD region. Overall, the results of this study will be valuable for facilitating local and regional policy-makers to propose appropriate strategies and effective control measures of VOCs and photochemical pollution in other regions of China, especially where O3 formation is VOC-limited. However, it is noteworthy that the above results were obtained based only on measurements taken over one month, and in a specific season (i.e., autumn), which may only represent the characteristics of photochemical pollution in autumn at a receptor site in the PRD region." For details, please refer to Line 2, Page 27 – Line 6, Page 28 in the revised manuscript.

The absence of biogenic VOC (e.g., isoprene) contributions to ozone formation is of concern (please see specific comments). Since the results were determined for a one-month autumn sampling period, additional discussion of the applicability of the findings to other seasons is merited. Readers will need some additional information to evaluate the accuracy of the box model simulations (please see specific comments). Reply: The reviewer's comment is highly appreciated. We agree with the reviewer and consider that the result extracted from the PMF model might not applicable to all seasons because it was based on a one-month autumn measurement. Detailed responses to the individual specific comment/suggestion are as follows.

Specific comments Please note that the "hundreds" digits of the line numbers are not

visible in the PDF, so I refer to both page number and line number throughout this review. Reply: Thanks for the reviewer's consideration. The format of line numbers has been revised according to ACP's guideline in the revised manuscript.

p. 7, l 32 − 34. The exclusion of biogenic VOC measurements as inputs to PMF is a limitation. Because the measurement period was October 22 to November 20, 2014, it is possible that biogenic emissions were much lower than would be typical of spring or summer and the exclusion of biogenic VOC concentrations from the PMF inputs may not affect the conclusions for this study period. However, there should be some acknowledgment that the results pertain to a fall study and may not represent other seasons. Reply: The reviewer's suggestion is highly appreciated. The mean concentrations of biogenic species (i.e., isoprene) during the sampling period was 150 ± 17 pptv, much lower than those observed in summer at other rural sites in the PRD region (∼250-1400 pptv) (Lau et al, 2010; Ding et al., 2012; Lu et al., 2012; Yuan et al., 2018). Furthermore, previous studies have reported that VOC abundance at the PRD region was mostly controlled by anthropogenic emissions, while controlling anthropogenic VOCs emissions seems to be more feasible than biogenic emissions (Tsui et al., 2009; Leung et al., 2010; HKEPD, 2012; Ling and Guo, 2014). As the aim of this study was to quantify the contributions of anthropogenic emissions to ambient VOCs and evaluate their contributions to photochemical $O_3$ formation at a receptor site of the PRD region, here we only used the anthropogenic VOCs as input for the PMF model (Barletta et al., 2008; Liu et al., 2008; Zheng et al., 2010; Lu et al., 2012; Tan et al., 2012). Moreover, we agree with the reviewer and consider that the result extracted from the PMF model might not applicable to all seasons because it was based on a one-month autumn measurement. To clarify it, the following text has been added in the manuscript: "......the results of this study will be valuable for facilitating local and regional policy-makers to propose appropriate strategies and effective control measures of VOCs and photochemical pollution in other regions of China, especially where $O_3$ formation is VOC-limited. However, it is noteworthy that the above results were obtained based only on measurements taken over one month, and in a specific

season (i.e., autumn), which may only represent the characteristics of photochemical pollution in autumn at a receptor site in the PRD region." For details, please refer to Lines 1-6, Page 28 in the revised manuscript.

References Barletta, B., Meinardi, S., Simpson, I.J., Zou, S.C., Rowland, F.S., Blake, D.R.: Ambient mixing ratios of nonmethane hydrocarbons (NMHCs) in two major urban centers of the Pearl River Delta (PRD) region: Guangzhou and Dongguan, Atmos. Environ., 42, 4393-4408, https://doi.org/10.1016/j.atmosenv.2008.01.028, 2008. Ding, X., Wang, X. M., Gao, B., Fu, X. X., He, Q. F., Zhao, X. Y, Yu, J. Z., and Zheng, M.: Tracer-based estimation of secondary organic carbon in the Pearl River Delta, south China, J. Geophys. Res., 117, D05313, httpp://doi.org/10.1029/2011JD016596, 2012. HKEPD (Hong Kong Protection Department): Integrated Data Analysis and Characterization of Photochemical Ozone in Hong Kong. http://www.epd-asg.gov.hk/english/report/aqr.html, 2012. Lau, A. K. H., Yuan, Z., Yu, J. Z., and Louie, P. K.: Source apportionment of ambient volatile organic compounds in Hong Kong, Sci. Total Environ., 408, 4138-4149, https://doi.org/10.1016/j.scitotenv.2010.05.025, 2010. Leung, D. Y. C., Wong, P., Cheung, B. K. H., and Guenther, A.: Improved land cover and emission factors for modeling biogenic volatile organic compounds emissions from Hong Kong, Atmos. Environ., 44, 1456-1468, https://doi.org/10.1016/j.atmosenv.2010.01.012, 2010. Ling, Z. H., and Guo, H.: Contribution of VOC sources to photochemical ozone formation and its control policy implication in Hong Kong, Environ. Sci. Pollut. Res., 38, 180–191, https://doi.org/10.1016/j.envsci.2013.12.004, 2014. Liu, Y., Shao, M., Lu, S., Chang, C. C., Wang, J. L., and Chen, G.: Volatile Organic Compound (VOC) measurements in the Pearl River Delta (PRD) region, China, Atmos. Chem. Phys., 8, 1531-1545, https://doi.org/10.5194/acp-8-1531-2008, 2008. Lu, K. D., Rohrer, F., Holland, F., Fuchs, H., Bohn, B., Brauers, T., Chang, C. C., Häseler, R., Hu, M., Kita, K., Kondo, Y., Li, X., Lou, S. R., Nehr, S., Shao, M., Zeng, L. M., Wahner, A., Zhang, Y. H., and Hofzumahaus, A.: Observation and modelling of OH and HO2 concentrations in the Pearl River Delta 2006: a missing OH source in a VOC rich atmosphere, Atmos. Chem.

Phys., 12, 1541-1569, https://doi.org/10.5194/acp-12-1541-2012, 2012. Tan, J. H., Guo, S. J., Ma, Y. L., Yang, F. M., He, K. B., Yu, Y. C., Wang, J. W., Shi, Z. B., and Chen, G. C.: Non-methane hydrocarbons and their ozone formation potentials in Foshan, China. Aerosol Air Qual. Res., 12, 387–398, http://doi.org/10.4209/aaqr.2011.08.0127, 2012. Tsui, J. K. Y., Guenther, A., Yip, W. K., Chen, F., A biogenic volatile organic compound emission inventory for Hong Kong, Atmos. Environ., 43, 6442-6448, https://doi.org/10.1016/j.atmosenv.2008.01.027, 2009. Yuan, J., Ling, Z. H., Wang, Z., Lu, X., Fan, S. J., He, Z. R., Guo, H., Wang, X. M., and Wang, N.: PAN–Precursor Relationship and Process Analysis of PAN Variations in the Pearl River Delta Region, Atmosphere, 9(10), 372, https://doi.org/10.3390/atmos9100372, 2018. Zheng, J.Y., Zheng, Z. Y., Yu Y. F., and Zhong, L. J.: Temporal, spatial characteristics and uncertainty of biogenic VOC emissions in the Pearl River Delta region, China. Atmos. Environ., 44, 1960-1969, https://doi.org/10.1016/j.atmosenv.2010.03.001, 2010.

p 8, | 52 – 54. Please clarify if biogenic VOCs were included among the inputs to the box model. The RIR values will not be correct if biogenic VOCs were excluded from the inputs to the box model. Reply: Thanks for the reviewer's comment. In this study, only one biogenic species (isoprene) was quantified at the Heshan site. We agreed with the reviewer that if the whole chunk of biogenic VOCs were excluded as inputs in photochemical box model, there will be bias for the model simulation. Actually, the simulation of PBM-MCM in this study included the observed concentrations of biogenic species (isoprene). To clarify that biogenic species, i.e., isoprene, was input into the PBM-MCM model, the text has been revised as follows: "In this study, the hourly data of VOCs, including both anthropogenic and biogenic species, five trace gases (i.e., $O_3$, NO, $NO_2$, CO, and $SO_2$) and two meteorological parameters (i.e., temperature and relative humidity) measured during the campaign were used as the model input." For details, please refer to Lines 17-19, Page 9 in the revised manuscript.

p 8, | 52 – 54. Please clarify that the model was applied to each individual day in the study for the RIR calculations (rather than that hourly data were averaged across days

and then input as averages to the model as was later done to generate the ozone iso-pleths). The citations to Lyu et al. (2016) and Wang et al. (2017) imply that the model was applied to each individual day (as done in those studies) for the RIR calculations but it would be helpful to be explicit in this paragraph. Reply: Thanks for the reviewer's great suggestion. Indeed, similar to Lyu et al. (2016), the PBM-MCM model was applied to the observed data collected in each individual day for the RIR calculation during the whole sampling period. To highlight the above model configuration, the following text has been added in the revised manuscript: "......Similar to Lyu et al. (2016), the PBM-MCM model was applied to the observed data collected on each individual day for the RIR calculation during the whole sampling period, while the hourly data during the whole sampling period were averaged across sampling days to provide mean diurnal variation as a base-case input for the PBM-MCM model to generate the O3 isopleths." For details, please refer to Lines 19-23, Page 9 in the revised manuscript.

Reference Lyu, X., Guo, H., Simpson, I. J., Meinardi, S., Louie, P. K. K., Ling, Z., Wang, Y., Liu, M., Luk, C. W. Y., Wang, N., and Blake, D. R.: Effectiveness of replacing catalytic converters in LPG-fueled vehicles in Hong Kong, Atmos. Chem. Phys., 16, 6609-6626, https://doi.org/10.5194/acp-16-6609-2016, 2016.

p 8, | 64. It might be helpful to say "net O3 production plus NO consumed" rather than "net O3 production and NO consumed" (the equations are correct). Reply: Thanks for the reviewer's suggestion. The text has been revised accordingly. For details, please refer to Line 7, Page 9 in the revised manuscript.

P 9, | 95 – 96. If O3 formation at the site is VOC-limited as previously discussed, it isn't likely that photochemical processing of the arriving air masses is truly complete. Even if NOx mixing ratios are too low to sustain further O3 production in the arriving air, injection of fresh NOx emissions into aged air masses would permit further pho-tochemical processing. The next section (3.2.1) indicates that photochemical aging was relatively incomplete (high correlations between more reactive and less reactive species; similarities between initial and observed VOC species concentrations except

for highly reactive species such as some alkenes and aromatics). Reply: Sorry for the mistake and confusion it caused. We agree with the reviewer that the air mass arrived at the site was not through complete photochemical processing. Furthermore, we also agreed with the reviewer #1's comment that variations of VOCs were related to the variations on sources other than photochemical processing. Therefore, the text has been deleted accordingly in the revised manuscript. For details, please refer to Line 24, Page 10 in the revised manuscript.

P 11, Table 1. It would be helpful to include summary statistics for NO and NO2. Reply: Thanks for reviewer's great suggestion. The summary statistics for NO and NO2 have been added in Table 1, and the following text has been added in the revised manuscript: "The mean mixing ratios of NO and NO2 were $4.2 \pm 0.4$ ppbv and $39.9 \pm 1.2$ ppbv, respectively, at the HS during the measurement, and the relatively low ratio of NO/NOx indicated that the site was distant from the source areas (Qin and Zhao, 2003; Melkonyan and Kuttler, 2012; Hagenbjörk et al., 2017)." For details, please refer to Lines 12-15, Page 10 in the revised manuscript.

Table 1. Average, range, and standard deviation of concentrations for NOx (i.e., NO and NO2) and the eight most abundant volatile organic compounds (VOCs) measured at the Heshan site, together with a sum of the mixing ratios for each hydrocarbon category (i.e., alkanes, aromatics, and alkenes). Species Average $\pm$ standard deviation (ppbv) Range (ppbv) NO $4.22 \pm 5.58$ 0.50 - 35.35 NO2 $39.92 \pm 16.12$ 8.02 - 130.77 Ethane $3.86 \pm 1.34$ 1.08 - 10.44 Toluene $3.74 \pm 2.89$ 0.56 - 15.80 acetylene $3.42 \pm 2.33$ 0.11 - 28.22 Propane $3.01 \pm 1.82$ 0.45 - 10.70 Ethene $2.94 \pm 3.34$ 0.37 - 64.56 i-Pentane $1.90 \pm 2.20$ 0.22 - 16.16 n-Butane $1.85 \pm 1.28$ 0.20 - 10.10 m/p-Xylene $1.62 \pm 1.44$ 0.17 - 12.74 Alkanes $17.04 \pm 10.64$ 2.04 - 61.01 Aromatics $9.07 \pm 7.01$ 1.46 - 40.12 Alkenes $5.29 \pm 5.01$ 0.67 - 77.39

P 17, section 3.3. For readers to understand the accuracy of the box model, it would be helpful to provide model performance statistics for the base case at the beginning of this section. Reply: Thanks for reviewer's helpful suggestion. The following text has been

added to evaluate the model performance: "To quantitatively evaluate the performance of the O3 simulation, the index of agreement (IOA), which was widely used for evaluation of PBM-MCM model (Wang et al., 2015; Wang et al., 2017; Liu et al, 2019) was introduced. The IOA was calculated by Eq. (7) (Huang et al., 2005): $IOA=1-(\sum\_(i=19n(O\_i-S\_i)2)/(\sum\_(i=19n(|O\_i-O|+|S\_i-O|)2),(7) where O i and S i represented observed and simulated concentrat...$ suggesting the abundance and variation of O3 were reasonably reproduced and could be used for further calculation." For details, please refer to Line 8, Page 19 – Line 5, Page 20 in the revised manuscript.

P 18, Figure 8 and related text. The RIR calculation for each species depends on the concentrations of all other species. If biogenic VOCs (e.g., isoprene) were not included as inputs to the box model, the reported RIR values will not represent the "real-world" condition. If biogenic VOCs were included as inputs to the box model, Figure 8 should show their RIRs and relative contributions. Please note in the caption what the error bars represent (e.g., one or two standard errors of the means or a different measure of variation?) Reply: The reviewer's comment is highly appreciated. We agreed with the reviewer that if the whole chunk of biogenic VOCs were excluded as inputs in photochemical box model, there will be bias for the model simulation due to it may not reflect the real atmospheric environment. In this study, only the most important biogenic species in this region, i.e., isoprene, was quantified at the Heshan site, which were actually included in the PBM-MCM model. In the revised manuscript, a discussion on the RIRs and relative contributions of VOC species have been added as follows, together with the description for the error bars: "Furthermore, Fig. 8c-f also showed the mean RIR values and the contributions to photochemical O3 formation for the top 10 VOC species and groups at the HS. Aromatics had the highest RIR value, with the average contribution of ∼82% to the sum RIR of all VOCs, followed by alkenes (∼11%) and alkanes (∼7%). Among the individual VOC species, toluene and m/p-xylene made most significant contribution (with a relative contribution of ∼40% and ∼34%, respectively) to O3 formation at the site when both the reactivity and abundance of VOC species were considered. The PMF results suggested that aromatics (including

toluene, xylenes, and ethylbenzene) were mainly from gasoline vehicular emission and solvent usage, while alkenes were mainly related to diesel vehicular emission. Overall, gasoline vehicular emission was the dominant contributor to O3 formation at the HS, and greater efforts should be devoted to toluene, xylenes, ethylbenzene, ethene, and 1-pentene for effectively controlling photochemical pollution." On the other hand, the error bars in Fig. 8 represented one standard errors of the mean RIR values. For details, please refer to Line 17, Page 20 – Line 7, Page 20 in the revised manuscript.

P 18, | 37 - 38. The phrase "based on the average diurnal variation" suggests that hourly data were averaged across days to provide a single base-case input for the box model's generation of ozone isopleths, which would be consistent with the approach described by Lyu et al. (2016). Please expand this description slightly. Do the 58 VOCs include isoprene and alpha- or beta-pinene? Reply: Thanks for the reviewer's comment. The description on PBM-MCM model running has been expanded in the revised manuscript. The simulation of PBM-MCM in this study indeed included the concentration of biogenic species (i.e., isoprene). If the whole chunk of biogenic VOCs were excluded as inputs in photochemical box model, there will be bias due to it cannot reflect the real atmospheric environment. It has been added or advised as follows: "In this study, the hourly data of VOCs, including both anthropogenic and biogenic species, five trace gases (i.e., O3, NO, NO2, CO, and SO2) and two meteorological parameters (i.e., temperature and relative humidity) measured during the campaign were used as the model input. Similar to Lyu et al. (2016), the PBM-MCM model was applied to the observed data collected on each individual day for the RIR calculation during the whole sampling period, while the hourly data during the whole sampling period were averaged across sampling days to provide mean diurnal variation as a base-case input for the PBM-MCM model to generate the O3 isopleths." "......The PBM-MCM model was employed based on the average hourly observed data across days to provide a single base-case input for the box model's generation of O3 isopleths. ......" For details, please refer to Lines 17-23, Page 9 and Lines 11-13, Page 21 in the revised manuscript.

P 19, | 61. Even in the NOx-limited regime, VOC reductions never increase ozone formation. Consistent with past studies, the isopleths are nearly vertical on the left side of Figure 9b and indicate a very small ozone decrease in response to VOC reductions at fixed NOx. Reply: Sorry for the mistake. It has been revised as follows: "On the other side of the ridge, for the reduced NOx to 7.5-15% of original mixing ratio, O3 concentration decreases with NOx concentration, but decreased VOCs would lead to minimal O3 variation." For details, please refer to Lines 17-19, Page 22 in the revised manuscript.

Pages 21 – 22. Many readers will not be able to follow this discussion as written. Reply: Thanks for the reviewer's patience and suggestion. It has been rewritten in the revised manuscript. For details, please refer to Line 7, Page 23 – Line 10, Page 26.

p. 23, | 34 – 35. Does this conclusion apply to other seasons? Reply: Thanks for the reviewer's comment. We consider that the results might not applicable to all seasons because it was based on a one-month autumn measurement. To clearly clarify that the results of this study was obtained based on the one-month measurement at autumn at a receptor site of the PRD region, the following text has been added in the manuscript: "……the results of this study will be valuable for facilitating local and regional policy-makers to propose appropriate strategies and effective control measures of VOCs and photochemical pollution in other regions of China, especially where O3 formation is VOC-limited. However, it is noteworthy that the above results were obtained based only on measurements taken over one month, and in a specific season (i.e., autumn), which may only represent the characteristics of photochemical pollution in autumn at a receptor site in the PRD region." For details, please refer to Lines 1-6, Page 28 in the manuscript.

Minor comments Abstract. The abstract uses a term ("abatement ratio") that is defined in the text (page 21), requiring a full page of explanation there. Another full page (p 22) is needed to define and discuss the abatement ratios of the individual source types. To aide readers of the abstract who will not have already read pages 21 - 22, I suggest

revising the following sentence: "Sensitivity analysis indicated that in order to prevent the increment of O3 concentration, the abatement ratios of the individual VOC source vs. NOx should be higher than 3.8, 4.6, 4.6, and 3.3, respectively, for diesel vehicular emission, solvent usage, biomass burning, and gasoline vehicular emission, respectively." A sentence such as the following appears to me to better convey the intended meaning: "Sensitivity analysis indicated that combined VOC and NOx emission controls would effectively reduce incremental O3 formation when the ratios of VOC-to-NOx emission reductions were higher than 3.8 for diesel vehicular emission, 4.6 for solvent usage, 4.6 for biomass burning, and 3.3 for gasoline vehicular emission." Reply: We thank the reviewer's helpful suggestion. It has been revised accordingly as follows: "Sensitivity analysis indicated that combined VOC and NOx emission controls would effectively reduce incremental O3 formation when the ratios of VOC-to-NOx emission reductions were > 3.8 for diesel vehicular emission, > 4.6 for solvent usage, > 4.6 for biomass burning, and 3.3 for gasoline vehicular emission." For details, please refer to Line 25, Page 1 – Line 3, Page 2 in the revised manuscript.

P 3, | 71-73. Please clarify if the percentages refer to percent of VOC mass or percent of VOC reactivity. Reply: Sorry for the mistake and confusion it caused. It has been revised as follows: "Results from source apportionment using the PMF model demonstrated the important roles of vehicular emissions in ambient VOCs in urban and suburban environments of Hong Kong, accounting for 48-54% and 31-40% of the concentrations of VOCs, respectively (Lau et al., 2010; Guo et al., 2011a)." For details, please refer to Line 24, Page 3 – Line 1, Page 4 in the revised manuscript.

P 5, | 99-100. The map shows that the site is southwest of Guangzhou and Foshan, not northeast (alternatively, the cities are northeast of the site). Reply: Sorry for the mistake. It has been revised accordingly in the revised manuscript (Line 3, Page 5).

P 13, eq. 6 and | 53 – 54. For consistency, please use either kVOC or kNMHC in both eq. 6 and the text. Reply: Thanks for pointing this out. It has been uniformly expressed as "kVOC" in the revised manuscript (Line 13, Page 14).

p. 20. Figure 9 is confusing for two reasons. First, it would be easier to understand the figure if the two panels were swapped (right replacing left). Second, the axes are also reversed from the more common presentation format. Reversing the vertical and horizontal axes and then placing the current left panel above the current right panel would be consistent with customary presentations (NOx reductions on the vertical, VOC reductions on the horizontal axis). Reply: Thanks for the great suggestion. It has been revised accordingly in the revised manuscript (Fig. 9, Page 23).

Please also note the supplement to this comment:
https://www.atmos-chem-phys-discuss.net/acp-2018-1293/acp-2018-1293-AC1-supplement.pdf

---

## Author Comment (AC2) · 20 May 2019

Response to Reviewers We appreciate the two anonymous reviewers for their constructive criticisms and valuable comments, which were of great help in improving the quality of the manuscript. We have revised the manuscript accordingly and our detailed responses are shown below. All the revision is highlighted in the revised manuscript.

Reviewer #1

The authors presented hourly resolved measurements of volatile organic compounds

(VOCs) at a receptor site in the Pearl River Delta (PRD) region in China. A receptor model, positive matrix factorization (PMF), was used to apportion sources of VOCs, by taking into account of photochemical degradation of some VOCs. Results showed that four sources, including gasoline vehicular emission, diesel vehicular emission, biomass burning, and solvent usage, are major contributors to anthropogenic VOCs at the site. A photochemical box model with the master chemical mechanism was also used to evaluate the contributions of those VOCs to ozone formation, with vehicular emission found to be the most significant. Further analysis to test abatement scenarios was also performed for policy implication. Overall, this is a nice data set with thorough analysis. The manuscript is generally well written and is surely of interest to readers of ACP. I therefore recommend minor revision before publication, with below comments for the authors. Reply: Thanks for the reviewer's positive comments and helpful suggestions. We have addressed all of the comments/suggestions in the revised manuscript. Detailed responses to the individual specific comment/suggestion are as follows.

Major: 1. I do not understand that in Section 3.3, why did the authors sum up gasoline and diesel vehicular emissions in their discussion on the contributions of different VOC sources to ozone formation. It is of importance for policy making, as to which type of vehicular to control (e.g., with priority), if we have more detailed understanding on whether vehicles run on gasoline or diesel have more potential in VOC emission that is related to ozone formation. Can the authors justify and clarify? Reply: The reviewer's comment is highly appreciated. We thought that both diesel and gasoline vehicles belong to vehicle, so we combined them as a whole to discuss the vehicular emission. We agree with the reviewer that more detailed understanding on whether diesel or gasoline vehicles contribute more to O3 formation is important for policy making. Therefore, we discuss different factors separately in the revised manuscript. The text has been revised as follows: "Figure 8a-b showed the mean RIR values of different VOC sources and NO, together with the contributions of different VOC sources to photochemical O3 formation. The mean RIR values of various VOC sources were positive, while that of NO was negative, suggesting that O3 formation at the HS was in the

VOC-limited regime. Among the four main anthropogenic sources of VOCs, relatively higher mean RIR values of vehicular emissions and biomass burning than that of solvent usage were found, with the mean RIR value of gasoline vehicular emission higher than that of diesel vehicular emission. Furthermore, considering both the reactivity and abundance of VOCs in different sources, the results showed that the gasoline vehicular emission was the most important contributor to photochemical $O_3$ production (Fig. 8b), with the mean percentage of 42%, followed by diesel vehicular emission (23%), biomass burning (20%) and solvent usage (15%), suggesting that controlling vehicular emissions (especially gasoline vehicular emission) and biomass burning could be a more effective way of reducing $O_3$ pollution in the region." For detail, please refer to Lines 6-16, Page 20 in the revised manuscript.

Figure 8. The mean RIR values of different sources (a) and their contributions to photochemical $O_3$ formation (b); The mean RIR values of different VOC groups (c) and their contributions to photochemical $O_3$ formation (d); The mean RIR values of top 10 VOCs (e) and their contributions to photochemical $O_3$ formation (f). The error bars represented one standard errors of the mean RIR values. Alkene* includes acetylene and alkenes except isoprene.

2. Using the same set of data, the same group of authors published in Journal of Environmental Science (JES) recently. Although the foci of the two papers are different, with the JES paper on isoprene and their oxidation products and this one on anthropogenic VOCs, I do like to see some connection between the two papers as they are based on the same data set (it was not even cited here). More importantly, what is the similarity and difference in methodology between these two papers? Would there be any bias if the whole chunk of biogenic VOCs were taken out from photochemical box model (e.g., the source/fate of OH radicals and ozone)? Reply: Thanks for the reviewer's comment. In the study published in JES (Ling et al., 2019), the source contributions of methacrolein (MACR) and methyl vinyl ketone (MVK) as well as their contributions to subsequently oxidation products were quantified. Both the study of

Ling et al. (2019) and this study applied PMF model for the source apportionment of VOCs based on the data collected at Heshan, but the aim of Ling et al. (2019) was to conduct the source apportionment of MACR and MVK, which included the primary emissions and secondary formation. Therefore, only species that are typical tracers of different emissions, including 18 NMHCs (i.e., isoprene, C6-C8 aromatics, C2-C4 alkenes), acetonitrile (ACN), methyl chloride (CH3Cl), methyl tert-butyl ether (MTBE) and peroxy acetyl nitrate (PAN) were selected as the input for the PMF model in Ling et al. (2019) (Song et al., 2006; Ho et al., 2009; Yuan et al., 2010, 2012b; Chen et al., 2014). On the other hand, different from that of Ling et al. (2019), this study aims to investigate and quantify the anthropogenic VOC sources which were all primary emissions. All anthropogenic VOCs except species with high uncertainties (i.e., cis-2-pentene, diphenyl methane, 1,3-diethylbenzene, etc. as more than a quarter of the samples for them were below the detection limits) were included in the PMF model. The total average concentration for the species in the PMF model accounted for ∼99% of that for all anthropogenic VOCs. Furthermore, as the fact that source apportionments of VOCs using the PMF model was conducted based on the assumption of mass conservation, a photochemical-aged-based parameterization was applied to identify the influence of photochemical processing on source signature of VOCs before running the PMF model in this study (Yuan et al., 2012b; Ling and Guo, 2014). Moreover, in this study, only one biogenic species (isoprene) was quantified at the Heshan site. We agreed with the reviewer that if the whole chunk of biogenic VOCs were excluded as input in the photochemical box model, there will be bias for the model simulation. Actually, the simulation of PBM-MCM in this study included the observed levels of biogenic species, i.e., isoprene. To clarify the similarities and difference between this and previous study (Ling et al., 2019), the text has been revised as follows: "The detailed description of the model input is provided elsewhere (Guo et al., 2011a; Ling et al., 2014). The selection of species for the PMF model followed the following principles: 1) the chosen species had relatively high concentrations and/or were typical tracers for specific emissions, e.g., methyl tert-butyl ether (MTBE) as the

tracer of gasoline vehicular exhaust (Song et al., 2006; Ho et al., 2009) and acetonitrile (ACN) as the tracer of biomass burning (Holzinger et al., 1999; Yuan et al., 2010); 2) species with low abundance and/or high uncertainties were excluded, i.e., cis-2-pentene, diphenyl methane, 1,3-diethylbenzene, etc., because more than a quarter of the samples for those species were below detection limits, and 3) species related to biogenic emissions (i.e., isoprene) were excluded as this study focused on the source characteristics of anthropogenic emissions in the PRD region (Fuentes et al., 1996; Sanadze, 2004; Zheng et al., 2010a; Zhang et al., 2012). A total of 49 species (including 47 non-methane hydrocarbons (NMHCs), MTBE, and ACN) were selected for the input data, which accounted for $\sim$99 % of the total concentration of all measured anthropogenic VOCs. This was different from our previous study (Ling et al., 2019), where only species that are typical tracers of different emissions, including 18 NMHCs (i.e., isoprene, C6-C8 aromatics, C2-C4 alkenes), acetonitrile (ACN), methyl chloride (CH3Cl), methyl tert-butyl ether (MTBE) and peroxy acetyl nitrate (PAN) were input into the PMF model for the contributions of primary emissions and secondary formation to ambient methacrolein (MACR) and methyl vinyl ketone (MVK) based on the same data set collected at the HS. For the PMF modelling, detailed information of the data processes and evaluation of the model performance has provided in previous studies (Lau et al., 2010; Ling et al., 2019). ……." To clarify that biogenic species, i.e., isoprene, was input into the PBM-MCM model, the text has been revised as follows: "In this study, the hourly data of VOCs, including both anthropogenic and biogenic species, five trace gases (i.e., O3, NO, NO2, CO, and SO2) and two meteorological parameters (i.e., temperature and relative humidity) measured during the campaign were used as the model input." For details, please refer to Line 17, Page 6 – Line 12, Page 7 and Lines 17-19, Page 9 in the revised manuscript.

References Chen, W. T., Shao, M., Lu, S. H., Wang, M., Zeng, L. M., Yuan, B., and Liu, Y.: Understanding primary and secondary sources of ambient carbonyl compounds in Beijing using the PMF model, Atmos. Chem. Phys., 14, 3047-3062, https://doi.org/10.5194/acp-14-3047-2014, 2014. Fuentes, J. D., Wang, D., Neumann,

H. H., Gillespie, T. J., Hartog, G. D., and Dann, T. F.: Ambient biogenic hydrocarbons and isoprene emissions from a mixed deciduous forest, J. Atmos. Chem., 25, 67-95, https://doi.org/10.1007/BF00053286, 1996. Guo, H., Zou, S. C., Tsai, W. Y., Chan, L. Y., and Blake, D. R.: Emission characteristics of non-methane hydrocarbons from private cars and taxis at different driving speeds in Hong Kong, Atmos. Environ., 45, 2711-2721, https://doi.org/10.1016/j.atmosenv.2011.02.053, 2011b. Guven, B. B., and Olaguer, E. P.: Ambient formaldehyde source attribution in Houston during TexAQS II and TRAMP, Atmos. Environ., 45(25), 4272-4280, https://doi.org/10.1016/j.atmosenv.2011.04.079, 2011. Ho, K. F., Lee, S. C., Ho, W. K., Blake, D. R., Cheng, Y., Li, Y. S., Ho, S. S. H., Fung, K., Louie, P. K. K., and Park, D.: Vehicular emission of volatile organic compounds (VOCs) from a tunnel study in Hong Kong, Atmos. Chem. Phys., 9, 7491-7504, https://doi.org/10.5194/acp-9-7491-2009, 2009. Lau, A. K. H., Yuan, Z., Yu, J. Z., and Louie, P. K.: Source apportionment of ambient volatile organic compounds in Hong Kong, Sci. Total Environ., 408, 4138-4149, https://doi.org/10.1016/j.scitotenv.2010.05.025, 2010. Ling, Z. H., and Guo, H.: Contribution of VOC sources to photochemical ozone formation and its control policy implication in Hong Kong, Environ. Sci. Pollut. Res., 38, 180–191, 2014. Ling, Z. H., He, Z. R., Wang, Z., Shao, M., and Wang, X. M.: Sources of MACR and MVK and their contributions to methylglyoxal and formaldehyde at a receptor site in Pearl River Delta, J. Environ. Sci., 79, 1-10, 2019. Ling, Z. H., Zhao, J., Fan, S. J., Wang, X. M.: Sources of formaldehyde and their contributions to photochemical O3 formation at an urban site in the Pearl River Delta, southern China, Chemosphere, 168, 1293-1301, 2017. Liu, Y., Shao, M., Fu, L., Lu, S., Zeng, L., and Tang, D.: Source profiles of volatile organic compounds (VOCs) measured in China: part I, Atmos. Environ., 42(25), 6247-6260. https://doi.org/10.1016/j.atmosenv.2008.01.070, 2008a. Lu, K. D., Rohrer, F., Holland, F., Fuchs, H., Bohn, B., Brauers, T., Chang, C. C., Häseler, R., Hu, M., Kita, K., Kondo, Y., Li, X., Lou, S. R., Nehr, S., Shao, M., Zeng, L. M., Wahner, A., Zhang, Y. H., and Hofzumahaus, A.: Observation and modelling of OH and HO2 concentrations in the Pearl River Delta 2006: a missing OH source in a VOC rich atmosphere, Atmos. Chem.

Phys., 12, 1541-1569, https://doi.org/10.5194/acp-12-1541-2012, 2012. Sanadze, G. A.: Biogenic Isoprene (A Review), Russian Journal of Plant Physiology, 51, 729-741, 2004. Song, C. L., Zhang, W. M., Pei, Y. Q., Fan, G. L. and Xu, G. P.: Comparative effects of MTBE and ethanol additions into gasoline on exhaust emissions, Atmos. Environ., 40, 1957-1970, https://doi.org/10.1016/j.atmosenv.2005.11.028, 2006. Yuan, B., Liu, Y., Shao, M., Lu, S., and Streets, D. G.: Biomass Burning Contributions to Ambient VOCs Species at a Receptor Site in the Pearl River Delta (PRD), China, Environ. Sci. Technol., 44, 4577, https://doi.org/10.1021/es1003389, 2010. Yuan, B., Shao, M., de Gouw, J., Parrish, D. D., Lu, S. H., Wang, M., Zeng, L. M., Zhang, Q., Song, Y., Zhang, J. B., and Hu, M.: Volatile organic compounds (VOCs) in urban air: How chemistry affects the interpretation of positive matrix factorization (PMF) analysis, J. Geophys. Res., 117, D24302, https://doi.org/10.1029/2012JD018236, 2012b. Zhang, Y.L., Wang, X.M., Blake, D.R., Li, L.F. Zhang, Z., Wang, S. Y., Guo, H., Lee, F. S. C., Gao, B., Chan, L. Y., Wu, D., Rowland, F. S.: Aromatic hydrocarbons as ozone precursors before and after outbreak of the 2008 financial crisis in the Pearl River Delta region, south China, J. Geophys. Res., 117, D15306, https://doi.org/10.1029/2011JD017356, 2012. Zheng, J.Y., Zheng, Z.Y., Yu, Y.F., and Zhong, L.J.: Temporal, spatial characteristics and uncertainty of biogenic VOC emissions in the Pearl River Delta region, China, Atmos. Environ., 44, 1960-1969, 2010.

3. Page 12, line 19-25. I do not understand the assertion here that correlation will be distorted if two VOCs from the same source have different photochemical reaction rates. The authors used the photochemical age concept in the paragraph right after it, which means that correction can still be retained if the two VOCs react in a proportional manner with OH radicals (and assuming no other fates). The rationale of these two paragraphs seems contradicting. Please clarify. Reply: Thanks for pointing it out. We agree with the reviewer that good correlation between two species from the same sources with different photochemical reaction rates would be retained if there were no other fates other than the oxidation by OH, NO3 and O3, as the fact that the reaction rates of these two VOCs are in a proportional manner. Therefore, the discussion on

the correlation between two species with different reaction rates has been deleted in the revised manuscript.

4. P7/L37: I would strongly suggest the authors include more details on why a four-factor solution was chosen. Diagnostic analysis by comparing three-factor and five-factor solutions would be useful, even as supplementary materials. "a good fit to the data and the most meaningful results" is just too descriptive and not very convincing. Reply: Thanks for the reviewer's suggestion. To clarify the determination of PMF solution, the following text has been added as follows: "In this study, the source apportionments of a 4-factor solution from the PMF model was selected, which were able to sufficiently and completely explain the levels and variations of observed VOCs (Lau et al., 2010) (Sect. 3.2.2). Compared with those of the 4-factor solution, the solution with 3 factors coerced two profiles that would otherwise be attributed to solvent usage and biomass burning, while certain amounts of aromatics and heptane were added into the profile of gasoline vehicular emissions. On the other hand, when the factor number was 5, an additional factor split from the biomass burning with the presence of C6-C9 alkanes, together with about 10-25 % of aromatics (including toluene and xylenes) found in the 5-factor solution. To evaluate the performance of the 4-factor solution, various tests and verifications were conducted. Firstly, different numbers of start seed in the model run were tested and it was found that there were no multiple solutions during the simulation. Furthermore, the scaled residuals of all the selected species ranged between -3 and 3 for the 4-factor solution, while the ratios of Q(robust)/Q(true) in this solution was close to 1 (Paatero, 2000a). In the 4-factor solution, strong correlations were found between the concentrations extracted from the model and the observed concentrations of each species, with correlation coefficients ranging from 0.71-0.95, indicating that the 4-factor solution well reproduced the observed variations of VOCs (Lau et al., 2010). In the bootstrapped simulation for the 4-factor solution, all the factors were mapped to a basic factor in all runs, indicating that the solution was stable. Finally, in the F-peak model results of the simulation, the G-space plot with no oblique edges suggested that the solution was with little rotation (Paatero, 2000a; USEPA, 2008). Overall, the above

features proved that the 4-factor solution from PMF could reliably attribute the sources of VOCs in this study." For details, please refer to Line 17, Page 7 – Line 12, Page 8 in the revised manuscript.

References: Lau, A.K.H., Yuan, Z.B., Yu, J.Z., Louie, P.K.K.: Source apportionment of ambient volatile organic compounds in Hong Kong, Sci. Total. Environ. 408, 4138-4149, 2010. Paatero, P.: User's guide for positive matrix factorization programs PMF2 and PMF3, part 1: tutorial, Prepared by University of Heisinki, Finland, February, 2000a. USEPA (U.S. Environmental Protection Agency): EPA Positive Matrix Factorization (PMF) 3.0 Fundamental and User Guide. July, 2008.

Minor: 1. P2/L41: "complex, nonlinear" to "complex and nonlinear". Reply: It has been revised accordingly.

2. P2/L44: "VOCs and NOx limited" to "VOC- and NOx-limited"; "VOCs-limited" to "VOC-limited", and in other places as well. Reply: Thanks for pointing this out. It has been revised accordingly in the revised manuscript. Furthermore, all "VOCs-limited" in the manuscript has been revised as "VOC-limited".

3. P3/L60: "emission-inventory" to "emission inventory". Reply: It has been revised accordingly in Line 12, Page 3 in the revised manuscript.

4. P4/L84: did Ling et al., JES, 2019 take photochemical processing into account? Reply: Thanks for the reviewer's comment. Our previous study (Ling et al., 2019) did not take the influence of photochemical processing on the variations of VOCs into account before using the PMF model as the study was to investigate the sources of MACR and MVK, which included primary and secondary sources. Furthermore, the species used in Ling et al. (2019) have relatively long lifetime, and it was suggested that the influence of photochemical processing were not significant on these species (Zhang et al., 2012), which could be further confirmed by Fig. 4 in our manuscript. For details, please refer to Line 1, Page 15 – Line 3, Page 16 in the revised manuscript.

References Ling, Z. H., He, Z. R., Wang, Z., Shao, M., and Wang, X. M.: Sources of MACR and MVK and their contributions to methylglyoxal and formaldehyde at a receptor site in Pearl River Delta, J. Environ. Sci., 79, 1-10, 2019 Zhang, Y. L., Wang, X. M., Blake, D. R., Li, L. F., Zhang, Z., Wang, S. Y., Guo, H., Lee, F. S. C., Gao, B., Chan, L. Y., Wu, D., and Rowland, F. S.: Aromatic hydrocarbons as ozone precursors before and after outbreak of the 2008 financial crisis in the Pearl River Delta region, south China, J. Geophys. Res.-Atmos., 117, D15306, https://doi.org/10.1029/2011JD017356, 2012.

5. P6/L19: remove "," after "contributions". Reply: Removed.

6. P7/L32: "the detection limit" to "their detection limits". Reply: Corrected. 7. P7/L35: why MTBE and ACN so special and not considered as "VOCs"? Reply: Sorry for the mistake. Indeed, MTBE and ACN were VOCs. It has been revised as follows: "A total of 49 species (including 47 non-methane hydrocarbons (NMHCs), MTBE, and ACN) were selected for the input data..." In addition, the following text has been added in the revised manuscript: "...the chosen species had relatively high concentrations and/or were typical tracers for specific emissions, e.g., methyl tert-butyl ether (MTBE) as the tracer of gasoline vehicular exhaust (Song et al., 2006; Ho et al., 2009) and acetonitrile (ACN) as the tracer of biomass burning (Holzinger et al., 1999; Yuan et al., 2010); ..." For details, please refer to Lines 3-4, Page 7 and Lines 18-21, Page 6 in the revised manuscript.

8. P8/L58: add "divided by" after "production"? Reply: Thanks for pointing this out. "divided by" has been added accordingly in the revised manuscript (Line 1, Page 9).

9. P9/L89&L91: "that" to "those". Reply: Corrected.

10. P9/L94-96: how can variations of VOCs suggest photochemical processing? It could be just variations on sources. Please clarify. Reply: The reviewer's comment is highly appreciated. We agreed with the reviewer that variations of VOCs were related to the variations on sources other than photochemical processing. Therefore, the text has been deleted accordingly in the revised manuscript. For details, please refer to

[Figure]

Line 24, Page 10 in the revised manuscript.

11. P9/L96: delete "to be". Reply: Deleted. 12. P12/L34-35: these rate constants appeared later in the next page and look redundant. Please remove. Reply: Thanks for pointing this out. Those redundant rate constants in Line 2, Page 14 have been deleted in the revised manuscript.

13. P13/Eq 6: $k_{VOC}$ instead of $k_{NMHC}$? It would be good to have a table showing the rate constants for each VOCs and appropriate citation. Reply: Thanks for pointing this out. It has been revised accordingly in the revised manuscript (Line 13, Page 14). In addition, a table showing the OH reaction rate constants for each VOC has been added as Table S1 in the supplement.

14. P14/L60&L62: "reaction rate" should be "reaction rate constant"? Reply: Corrected.

15. P14/L71: add "the" before "rest species". Reply: Added

16. P15/L89: remove "," after "VOCs". Reply: Removed.

17. P15/L93: "acetonitrile" to "ACN" (you defined it early). Reply: Thanks for pointing this out. It has been revised accordingly in the revised manuscript (Line 1, Page 17).

18. P15/L00: "peak" or "valley"? Reply: Sorry for the mistake. It has been corrected to "valley" in the revised manuscript (Line 10, Page 18).

19. P17/L22: "relative" to "relatively". Reply: Corrected. 20. P18/L29: add "a" before "more". Reply: It has been revised accordingly.

21. P23/L31: remove "cluster". Reply: Removed.

22. P24/L44-P25/L65: I would suggest to shorten this paragraph to a few sentences to make the point: although there are some control measures on VOC emission from vehicles, there is limited control on biomass burning etc. Reply: Thanks for the great suggestion. To condense this paragraph and to make brief description on the policy

implication, the text has been revised as follows: "Indeed, many additional policies on VOCs have been and continue to be implemented and formulated in the PRD region. A series of policies regarding the control of vehicular emission have been conducted in the PRD region, the purposes of which can be mainly divided into two categories: 1) improve the environmental standards of the main air pollutants and standards of emissions; and 2) improve the quality of the fuel used in vehicles. Policies on controlling biomass burning, however, are relatively limited. Nevertheless, some policies have been effective, and levels of NOx (another important O3 precursor), have decreased in the PRD region in recent years. On the other hand, for VOCs, most relevant policies only control the total mass and/or the total emissions of VOCs, and the level of O3 continues to increase in this region. ……." For details, please refer to Lines 13-21, Page 27 in the revised manuscript.

23. Figure 4: can the authors use another panel to show ratios too? Reply: Thanks for pointing this out. Figure 4 has been revised to show the ratios of initial to observed concentrations of VOCs in the revised manuscript. (Fig. 4, Page 15).

Figure 4. Ratios of initial to observed concentrations of volatile organic compounds (VOCs).

24. Figures in general: it would be more reader friendly if the authors can use a bigger font size for most of the figures. Reply: The reviewer's suggestion is highly appreciated. The font size for most of the figures have been bigger in the revised manuscript.  

Please also note the supplement to this comment:
https://www.atmos-chem-phys-discuss.net/acp-2018-1293/acp-2018-1293-AC2-supplement.pdf

---

## Author Comment (AC3) · 20 May 2019

Response to Reviewers We appreciate the two anonymous reviewers for their constructive criticisms and valuable comments, which were of great help in improving the quality of the manuscript. We have revised the manuscript accordingly and our detailed responses are shown below. All the revision is highlighted in the revised manuscript.

Reviewer #2

This manuscript presents an analysis of hourly VOC data from the Pearl River Delta re-
gion including PMF and VOC sources important for ozone formation. There is also an analysis of VOC and NOx limitation and what effect controlling each of these may have on ozone. In general the manuscript is well written with only a few sections needing clarification. Reply: Thanks for the reviewer's positive comments and helpful suggestions. We have addressed all of the comments/suggestions in the revised manuscript. Detailed responses to the individual specific comment/suggestion are as follows.

Specific Comments: P8 Line 62-63: Is this a result from this study or from elsewhere? It is confusing to have these papers cited without more explanation. Reply: Sorry for the mistake and confusion it caused. We have revised this part to clarify it, and the improved text is as follows: "As ethane, acetylene, ethene, and propane had been suggested to be the tracers of incomplete combustion from vehicle exhaust, biomass or coal (Liu et al., 2008a; Lau et al., 2010; Guo et al., 2011b; Yuan et al., 2012a), the characteristic of abundant species at the HS indicated that incomplete combustion was likely the dominant source of VOCs during the measurement period." For details, please refer to Line 24, Page 10 – Line 4, Page 11 in the revised manuscript.

Table 1: Add a line dividing the individual species from the categories. This will make the table easier to read. Reply: Thanks for the reviewer's great suggestions. It has been revised accordingly in the revised manuscript (Table 1, Page 12).

P9 Line 73-78: The use of these correlations seems to be overstated both in terms of widespread use (1 paper cited) and what these correlations mean for photochemistry impacts. Weak correlations may exist due to strength of impact from different sources in addition to changing photochemistry. Adding some statements about the degree of correlation or lack of correlation between other species would strengthen this section. A more convincing argument would be to look at photochemically formed species from these precursor compounds – I'm not sure if these compounds are available. This simple analysis and the extended analysis using the parameterization suggests little photochemistry is taking place. Is this due to the time available for photochemistry to take place or atmospheric conditions. What is the estimated air mass age?

(using chemical tracers, proximity to sources and average wind speed, or from the parameterization). Reply: Thanks for the reviewer's comment. We agree with the reviewer that weak correlation between two species may exist due to different impact on them from different sources. Furthermore, the reviewer#1 pointed out that good correlation between two species from the same sources with different photochemical reaction rates would be retained if there were no other fates other than the oxidation by OH, NO3 and O3, as the fact that the reaction rates of these two VOCs are in a proportional manner. We agreed with both reviewers, and the discussion on the correlation between two species with different reaction rates has been deleted in the revised manuscript. In this study, we applied the photochemical-aged-based parameterization method to estimate the initial concentrations of VOCs after emissions. The OH exposure ([OH] $\Delta t$) is calculated and used to represent photochemical age, as [OH] and $\Delta t$ always appear together in the parameterization equation (Shao et al., 2009; Yuan et al., 2012b). $[OH]\Delta t=1/((k\_E-k\_X) )\times[\ln [E]/[X] |\_(t=0)-\ln [E]/[X] ]$ (5) The OH exposure ([OH]$\Delta t$) was calculated as $6.47\times109$ moleculeÂůcm-3Âůs. With the hourly concentrations of OH during the measurement period simulated by PBM-MCM, the air mass age $\Delta t$ was calculated as about 3 hours. As the lifetimes of most of species in this study ranged from 2.8 hours to 6 months (Simpson et al., 2010), the air mass age calculated by the parameterization method suggested that there was not enough time for these species to be degraded completely by OH radical. Indeed, from the parameterization method, the difference between initial and observed concentrations were small for most of the VOC species, i.e., the species with the OH reaction rate constant < $5.64\times10-11$ cm3Âůmolecule-1Âůs-1, and the ratio of initial/observed concentrations ranging from 1.00-1.23. For those species with relatively higher photochemical reactivity (with the OH reaction rate constant ranging from $5.64\times10-11-6.40\times10-11$ cm3Âůmolecule-1Âůs-1), the initial concentrations were 1.44-1.51 times the observed levels (Fig. 4). It should be noted that these relatively higher reactive species only accounted for a small fraction of the concentrations and the ozone formation potential (OFP) of all the observed VOCs due to their relatively lower abundance (data not

shown). Furthermore, to roughly identify the potential source areas of VOCs, 24-h air masses backward trajectories in 3-h intervals were calculated, and 7 main types of backward trajectories were obtained through cluster analysis (Fig. S2), which were mostly passing through the center cities of PRD before arriving at the Heshan site. As the air mass age was calculated as about 3 hours by the photochemical-aged-based parameterization method, the position of 3-h backward trajectories of each type was extracted to determine the source areas. It was found that ∼70% air masses were from the center cities of PRD (i.e., Foshan and Guangzhou), while ∼30% were from the southeast of Jiangmen city and from the center of Zhongshan city, indicating that the air masses at the Heshan site were from or through the urban areas with significant anthropogenic emissions. The above discussion was provided in the revised manuscript (Lines 8-14, Page 14) and the supplementary.

Figure S2. The seven clusters of 24-h air masses backward trajectories with Heshan as the ending point (the trajectories were simulated for 3-h intervals at the ending point of 200 m above sea level) (Ling et al., 2013).

References: Ling, Z.H., Guo, H., Zheng, J.Y., et al.: Establishing a conceptual model for photochemical ozone pollution in subtropical Hong Kong. Atmos. Environ. 76, 208-220, 2013. Shao, M., Lu, S.H., Liu, Y., Xie, X., Chang, C.C., Huang, S., and Chen, Z.M.: Volatile organic compounds measured in summer in Beijing and their role in ground-level ozone formation, J. Geophys. Res., 114, D00G06, https://doi.org/10.1029/2008JD010863, 2009. Simpson, I.J., Blake, N.J., Barletta, B., et al.: Characterization of trace gases measured over Alberta oil sands mining operations: 76 speciated C2-C10 volatile organic compounds (VOCs), CO2, CH4, CO, NO, NO2, NOy, O3 and SO2, Atmos. Chem. Phys., 10.11931-11954, 2010. Yuan, B., Chen, W.T., Shao, M., Wang, M., Lu, S.H., Wang, B., Liu, Y., Chang, C.C., Wang, B.G.: Measurements of ambient hydrocarbons and carbonyls in the Pearl River Delta (PRD), China, Atmos. Res., 116, 93-104, https://doi.org/10.1016/j.atmosres.2012.03.006, 2012a.

Figure 2: Consider showing as the difference between the measured and estimated initial concentration instead. It will be easier to see the delta for all species than infer it from the difference between bars. Reply: Thanks for the reviewer's great suggestions. Figure 4 was revised to show the ratios of initial to observed concentrations of VOCs in the revised manuscript (Fig. 4, Page 15).

Figure 4. Ratios of initial to observed concentrations of volatile organic compounds (VOCs).

Figure 5: Why are both diesel and gasoline vehicles together if they were different factors? Reply: The reviewer's comment is highly appreciated. The similar comment was also made by reviewer #1. We agree with the reviewer that more detailed understanding on whether diesel or gasoline vehicles contribute more to O3 formation is important for policy making. Therefore, we discuss different factors separately in the revised manuscript. The text has been revised as follows: "Figure 8a-b showed the mean RIR values of different VOC sources and NO, together with the contributions of different VOC sources to photochemical O3 formation. The mean RIR values of various VOC sources were positive, while that of NO was negative, suggesting that O3 formation at the HS was in the VOC-limited regime. Among the four main anthropogenic sources of VOCs, relatively higher mean RIR values of vehicular emissions and biomass burning than that of solvent usage were found, with the mean RIR value of gasoline vehicular emission higher than that of diesel vehicular emission. Furthermore, considering both the reactivity and abundance of VOCs in different sources, the results showed that the gasoline vehicular emission was the most important contributor to photochemical O3 production (Fig. 8b), with the mean percentage of 42%, followed by diesel vehicular emission (23%), biomass burning (20%) and solvent usage (15%), suggesting that controlling vehicular emissions (especially gasoline vehicular emission) and biomass burning could be a more effective way of reducing O3 pollution in the region." For detail, please refer to Lines 6-16, Page 20 in the revised manuscript.

Figure 8. The mean RIR values of different sources (a) and their contributions to photochemical O3 formation (b); The mean RIR values of different VOC groups (c) and their contributions to photochemical O3 formation (d); The mean RIR values of top 10 VOCs (e) and their contributions to photochemical O3 formation (f). The error bars represented one standard errors of the mean RIR values. Alkene* includes acetylenes and alkenes except isoprene.

Sect 3.3 How does this compare for the different classes of VOCs? This would be good to include to inform which measurements would be most important to make long-term or in the future. Or if new industry moved into the area and the mix of VOCs was different. Reply: The reviewer's suggestion is highly appreciated. The relevant discussion on different VOC species and groups has been added in the revised manuscript as follows: "Furthermore, Fig. 8c-f also showed the mean RIR values and the contributions to photochemical O3 formation for the top 10 VOC species and groups at the HS. Aromatics had the highest RIR value, with the average contribution of ∼82% to the sum RIR of all VOCs, followed by alkenes (∼11%) and alkanes (∼7%). Among the individual VOC species, toluene and m/p-xylene made most significant contribution (with a relative contribution of ∼40% and ∼34%, respectively) to O3 formation at the site when both the reactivity and abundance of VOC species were considered. The PMF results suggested that aromatics (including toluene, xylenes, and ethylbenzene) were mainly from gasoline vehicular emission and solvent usage, while alkenes were mainly related to diesel vehicular emission. The results suggested that gasoline vehicular emission was the dominant contributor to O3 formation at the HS, and greater efforts should be devoted to toluene, xylenes, ethylbenzene, ethene, and 1-pentene for effectively controlling photochemical pollution." For detail, please refer to Line 17, Page 20 – Line 7, Page 21 in the revised manuscript.

Sect 3.4.2 – something about the tenses used (past and present) is confusing or not consistent. I suggest going through the section and checking verb use for consistency and clarity. Reply: Thanks for the reviewer's patience and revision. For better expression, we have rewritten section 3.4.2 and double-checked the whole manuscript.

Line 19 Line 40: Which of your reduction scenarios reflect these changes? I ask that for the majority of this section. The listing of all these policies without more direct connection to your findings is superfluous. Only the last paragraph in the section touches on this but still doesn't connect how much these policies are expected to change VOC and NOx levels. The importance and relevance of this section needs to be considered. If it is important figure out a way to make it easier to follow and more connected to the rest of the paper. Reply: The reviewer's suggestion is highly appreciated. In this study, we summarize briefly the policy for controlling VOC emissions conducted/being conducted in the PRD region based on the source apportionments of anthropogenic VOCs and their contributions to O3 formation at a receptor site of the PRD region. Though the results, i.e., the source apportionments of VOCs, their contributions to O3 formation and the appropriate reduction ratios, indeed provided scientific support and indication for devising controlling strategies on photochemical pollution in this region, this study could not evaluate the benefits and dis-benefits of the control measures, i.e., how much the policies influence on the abundance of VOCs and NOx, only by one-month measurement data collected in a specific season. Therefore, to shorten the summary of the policy as suggested by reviewer #1 and to connect the results of this study to the implication for policy development, the policy implication section has been combined with the conclusion section and revised as follows: "The PRD region has long been facing severe photochemical air pollution, and VOCs has been the limiting factor of O3 formation in this region. To better understand the contribution of different anthropogenic VOCs to O3 formation in this region, we performed in-depth analyses on data obtained from intensive measurements of VOCs and related species at a downwind rural site (Heshan site, HS) of the PRD region during October – November, 2014. Four anthropogenic sources were identified by the PMF model with the consideration of the influence of photochemical processing. The O3 formation at the HS was generally VOC-limited, with the vehicular emission (especially gasoline vehicular emission) as the most important anthropogenic VOC source contributing to O3 formation, followed by biomass burning. It indicated that priority should be given to controlling vehicular

emission and biomass burning. Furthermore, with the current industries operating in the PRD region, particular attention should be given to toluene, xylenes, ethylbenzene, ethene and 1-pentene in efforts to control photochemical pollution. Indeed, many additional policies on VOCs have been and continue to be implemented and formulated in the PRD region. A series of policies regarding the control of vehicular emission have been conducted in the PRD region, the purposes of which can be mainly divided into two categories: 1) improve the environmental standards of the main air pollutants and standards of emissions; and 2) improve the quality of the fuel used in vehicles. Policies on controlling biomass burning, however, are relatively limited. Nevertheless, some policies have been effective, and levels of NOx (another important O3 precursor), have decreased in the PRD region in recent years. On the other hand, for VOCs, most relevant policies only control the total mass and/or the total emissions of VOCs, and the level of O3 continues to increase in this region. To prevent net O3 increment, the VOCs and NOx should be controlled in an appropriate ratio since VOCs and NOx are frequently controlled simultaneously. Furthermore, long-term monitoring is still needed to evaluate the benefits and dis-benefits of the control measures on vehicular emissions and/or photochemical pollution in the PRD region. Overall, the results of this study will be valuable for facilitating local and regional policy-makers to propose appropriate strategies and effective control measures of VOCs and photochemical pollution in other regions of China, especially where O3 formation is VOC-limited. However, it is noteworthy that the above results were obtained based only on measurements taken over one month, and in a specific season (i.e., autumn), which may only represent the characteristics of photochemical pollution in autumn at a receptor site in the PRD region." For details, please refer to Line 2, Page 27 – Line 6, Page 28 in the revised manuscript.

P9 Line 77: little instead of insignificant. Insignificant implies statistics were used. Reply: Thanks for pointing this out. The relevant text has been deleted in the revised manuscript.

P11 Line 11: adjusted instead of compensated Reply: Thanks for pointing this out. It has been revised accordingly in the revised manuscript (Line 16, Page 15).

P12 Line 26: C6-C7 alkanes? Reply: Sorry for the mistake. It has been revised accordingly in the revised manuscript (Line 17, Page 16). P13 Line 39: why "again"? I don't think you've presented PBM-MCM data yet. Reply: Thanks for pointing this out. It has been deleted in the last manuscript.

P19: Line 32-33: This suggests...in the region. Reply: Thanks for the reviewer's suggestion. The relevant passage had been deleted in the last manuscript.

P19 Line 42: closely instead of strictly Reply: Thanks for the reviewer's suggestion. The relevant passage had been deleted in the last manuscript.

P19 Line 42: What are new energy automobiles? Reply: [P24 54] Sorry for the mistake and confusion it caused. The relevant passage had been deleted in the revised manuscript to shorten the policy implication section as suggested by the reviewer#1. The accurate expression should be "new energy vehicles". The term new energy vehicles (NEVs) is used to designate plug-in electric vehicles eligible for public subsidies in China, and includes only battery electric vehicles (BEVs), plug-in hybrid electric vehicles (PHEVs), and fuel cell electric vehicles (FCEV). (https://en.wikipedia.org)  

Please also note the supplement to this comment:
https://www.atmos-chem-phys-discuss.net/acp-2018-1293/acp-2018-1293-AC3-supplement.pdf